# PriSM: Prior-Guided Search Methods for Query Efficient Black-Box Attacks

**Pavlos Ntais**                                                                    *pntais@di.uoa.gr*
*Department of Informatics and Telecommunications*
*University of Athens*

**Thanassis Avgerinos**                                                        *thanassis@di.uoa.gr*
*Department of Informatics and Telecommunications*
*University of Athens*

**Reviewed on OpenReview:** *https://openreview.net/forum?id=UQsOh2kfhP*

## Abstract

Deep Neural Networks are vulnerable to adversarial examples in black-box settings, requiring query-efficient attack methods. We propose **PriSM** (**Pri**or-Guided **S**earch **M**ethods), which systematically exploits two types of transferable surrogate information: decision boundary geometry and loss landscape topography. We demonstrate their utility through complementary attacks: (1) **TGEA** leverages boundary geometry to initialize evolutionary optimization with surrogate evolved populations, maximizing attack success rates, and (2) **SGSA** leverages loss topography via multi-scale saliency guidance to direct Square Attack's (Andriushchenko et al., 2020) perturbations, minimizing query costs. Across MNIST, CIFAR-10, and ImageNet, both methods achieve 30-60% query reductions compared to uninformed baselines, while also being competitive with state of the art hybrid attacks. Our evaluation reveals a strategic trade off: SGSA excels in query efficiency through local exploitation, whereas TGEA maximizes success rates via global exploration. Our comprehensive evaluation also demonstrates that different types of surrogate information require matched exploitation strategies, providing practical guidance for query-efficient black-box attacks.

## 1 Introduction

Deep Neural Networks (DNNs) have achieved remarkable success across critical domains, including autonomous vehicles, biometric authentication, and medical diagnostics (Krizhevsky et al., 2012; He et al., 2016). However, their susceptibility to adversarial examples—subtly perturbed inputs that cause misclassification, poses significant security risks (Szegedy et al., 2014; Goodfellow et al., 2015; Carlini & Wagner, 2017).

In real-world deployments, attackers often operate under black-box constraints, where they lack access to model internals and can only query the model for predictions (Papernot et al., 2017). This setting presents two key challenges: (1) achieving high attack success rates without gradient information, and (2) minimizing the number of queries to evade detection and reduce computational costs (Chen et al., 2017; Ilyas et al., 2018). Existing approaches: transfer-based, query-based, and hybrid methods face a fundamental trade-off: transfer attacks require zero queries but suffer from low success rates, while query-based attacks achieve high success at the cost of thousands of queries per sample (Papernot et al., 2017; Dong et al., 2018; Chen et al., 2017; Andriushchenko et al., 2020; Huang & Yu, 2022). Opportunities remain to more fully leverage surrogate model information to enhance both query efficiency and attack success.

In this work, we introduce **PriSM** (Prior-Guided Search Methods), a novel framework that strategically incorporates surrogate-derived guidance in different ways in order to enhance black-box adversarial searches more effectively. Our key contributions are:

- **TGEA**: A transfer-guided evolutionary attack with advanced initialization strategies (TASI and SEGI) that warm-start CMA-ES optimization (Hansen, 2006), delivering the highest success rates on complex models.

- **SGSA**: A saliency-guided enhancement to the Square Attack (Andriushchenko et al., 2020) that uses multi-scale gradient maps from surrogates to intelligently inform perturbation placement and sizing, achieving state-of-the-art query efficiency.

- **Intuitive Analysis**: Intuitive justifications showing that SGSA exploits local loss geometry transferability, while TGEA leverages global decision boundary correlations.

Our results demonstrate that PriSM offers a strategic choice between minimizing query costs and maximizing attack success. Both SGSA and TGEA reduce queries by up to 50% over random baselines while achieving equal or greater success rates. This work advances the understanding of prior-guided paradigms in adversarial machine learning, providing practical tools for robustness evaluation.

## 2 Related Work

Black-box adversarial attacks balance success rates and query costs through three paradigms: transfer-based, query-based, and hybrid approaches, with recent advancements incorporating saliency guidance and evolutionary optimization.

Transfer-based attacks exploit cross-model transferability by generating perturbations on surrogate models trained on similar data distributions (Papernot et al., 2017; Liu et al., 2017). Efforts to boost transferability include momentum integration (Dong et al., 2018), input diversity (Xie et al., 2019), and ensemble methods (Liu et al., 2017). Although these require no queries to the target, their effectiveness is often limited against robust defenses (He et al., 2017).

Query-based attacks iteratively refine perturbations via direct model queries. Score-based methods like SimBA (Guo et al., 2019) and Square Attack (Andriushchenko et al., 2020) use confidence scores, with Square Attack leading benchmarks such as BlackboxBench (Zheng et al., 2025). Decision-based approaches like Boundary Attack (Brendel et al., 2018) rely solely on hard labels but incur higher query demands (Chen & Gu, 2020). Saliency-guided variants prioritize perturbations in salient regions (Dai et al., 2023; Soor et al., 2025).

Hybrid attacks combine these paradigms, using surrogate priors to initialize or guide query-based searches. Examples include TAGA (Huang & Yu, 2022), which refines transfer seeds via genetic algorithms, and Hybrid Batch Attacks (Suya et al., 2020), which leverage local surrogates for scalable generation. Recent work enhances transferability through meta-learning (Fu et al., 2022), ensemble surrogates (Cai et al., 2022), and saliency integration (Wang et al., 2024; Huang & Kong, 2022). Most recently, PBO (Cheng et al., 2024) introduced Bayesian optimization with a learned function prior, achieving state-of-the-art query efficiency on standard models by exploiting loss surface smoothness.

Evolutionary algorithms navigate high dimensional spaces without gradients (Su et al., 2019; Alzantot et al., 2019). CMA-ES (Hansen, 2006) adapts search distributions via covariance updates, showing promise in adversarial contexts, though random initializations can lead to slow convergence (Qiu et al., 2021; Kuang et al., 2019).

Despite these advances, existing hybrids often underutilize surrogate priors for continuous guidance or sophisticated warm-starts. Our PriSM framework addresses this through SGSA, which embeds multi-scale saliency from surrogates into every Square Attack query decision, and TGEA, which enhances evolutionary hybrids with surrogate-evolved initializations (e.g., SEGI), achieving up to 8.24% higher success rates on robust models while maintaining low query costs.

## 3 Background

### 3.1 Adversarial Attacks

Let $f_\theta : \mathcal{X} \to \mathbb{R}^K$ denote a deep neural network classifier with parameters $\theta$, mapping input $x \in \mathcal{X}$ to logits over $K$ classes. For a clean input $x$ with true label $y$, the model's prediction is $\hat{y} = \arg\max_k f_\theta(x)_k$.

An adversarial example $x' = x + \delta$ is a perturbed input that causes misclassification while remaining perceptually similar to $x$. The perturbation $\delta$ is typically constrained by $\|\delta\|_p \leq \epsilon$, where $p \in \{0, 2, \infty\}$ and $\epsilon$ controls the perturbation budget.

**White-box attacks.** With full access to model parameters and gradients, white-box attacks like PGD (Madry et al., 2019) iteratively optimize:

$$x^{t+1} = \Pi_\epsilon(x^t + \alpha \cdot \text{sign}(\nabla_x \mathcal{L}(f_\theta(x^t), y))) \tag{1}$$

where $\Pi_\epsilon$ projects onto the $\ell_p$ ball of radius $\epsilon$, and $\mathcal{L}$ is the attack loss (e.g., cross-entropy).

**Black-box attacks.** In black-box settings, attackers lack gradient access and can only query the model. These attacks are categorized by query type:

- **Score-based:** Access to confidence scores/logits
- **Decision-based:** Access only to predicted labels
- **Transfer-based:** Zero queries, relying on adversarial transferability

### 3.2 Square Attack

Square Attack (Andriushchenko et al., 2020) is a score-based black-box attack that does not rely on local gradient information. It selects localized square-shaped updates at random positions such that the perturbation is situated approximately at the boundary of the feasible set. At iteration $i$, it *randomly* samples a square region depending on the current square size $h^{(i)}$. The algorithm minimizes the margin-based loss $L(f(\hat{x}), y) = f_y(\hat{x}) - \max_{k \neq y} f_k(\hat{x})$. The update rule for the perturbation $\delta$ is defined as:

$$\delta^{i+1} = \begin{cases} \text{Project}(\delta^i + \nu_i) & \text{if } L(f_\theta(x + \text{Project}(\delta^i + \nu_i)), y) < L(f_\theta(x + \delta^i), y) \\ \delta^i & \text{otherwise} \end{cases} \tag{2}$$

where the update $\nu_i$ is sampled from a discrete distribution $\nu \in \{-2\epsilon, 2\epsilon\}^d$ within the square region, ensuring the perturbation stays at the corners of the $\ell_\infty$-ball. The **Project** function ensures the adversarial example remains within the valid $\epsilon$-ball bound:

$$\text{Project}(z) = \min(\max(z, x - \epsilon, 0), x + \epsilon, 1) \tag{3}$$

This operation constrains the perturbed input to the intersection $\{z \in \mathbb{R}^d : \|z - x\|_\infty \leq \epsilon\} \cap [0, 1]^d$

### 3.3 CMA-ES for Black-Box Optimization

Covariance Matrix Adaptation Evolution Strategy (CMA-ES) (Hansen, 2006) is a gradient-free optimizer that adapts a multivariate normal search distribution. At generation $g$, it samples $\lambda$ candidate solutions:

$$x_i^{(g)} \sim \mathcal{N}(m^{(g)}, (\sigma^{(g)})^2 C^{(g)}), \quad i = 1, \ldots, \lambda \tag{4}$$

where $m^{(g)}$ is the mean vector, $\sigma^{(g)}$ is the step size, and $C^{(g)}$ is the covariance matrix.

The mean is updated using the $\mu$ best individuals:

$$m^{(g+1)} = \sum_{i=1}^{\mu} w_i x_{i:\lambda}^{(g)} \tag{5}$$

where $x_{i:\lambda}$ denotes the $i$-th best solution and $w_i$ are recombination weights.

The covariance matrix adapts to the local loss landscape through rank-one and rank-$\mu$ updates, enabling efficient search in high-dimensional spaces without gradient information. However, CMA-ES suffers from slow convergence when initialized randomly, particularly for adversarial attack objectives.

### 3.4 Surrogate Models and Transferability

A surrogate model $f_s$ is a white-box model accessible to the attacker, typically trained on similar data as the target model $f_t$. Adversarial transferability refers to the phenomenon where adversarial examples crafted on $f_s$ often fool $f_t$:

$$\Pr[f_t(x + \delta_s) \neq y] > 0 \quad \text{where } \delta_s = \underset{\|\delta\|_p \leq \epsilon}{\arg\max} \, \mathcal{L}(f_s(x + \delta), y) \tag{6}$$

Transferability arises from models learning similar decision boundaries on similar data distributions (Papernot et al., 2017). However, transfer success varies significantly with model architecture, training procedure, and robustness (Huang et al., 2017).

**Saliency maps.** Gradient-based saliency maps indicate input region importance:

$$S(x) = \|\nabla_x \mathcal{L}(f_s(x), y)\|_2 \tag{7}$$

High saliency regions are more sensitive to perturbations. While saliency is model-specific, relative importance patterns often transfer across models (Simonyan & Zisserman, 2015), enabling surrogate-guided attacks.

## 4 Methodology

### 4.1 Motivation

In this work, we investigate how surrogate model information can be systematically exploited to improve query efficiency in black-box adversarial attacks. While transfer-based attacks require zero queries but suffer from low success rates, and query-based attacks achieve high success at the cost of thousands of queries, hybrid approaches that *strategically* leverage surrogate priors remain underexplored. To address this gap, we propose two methods that demonstrate the importance of matching surrogate information types to compatible search algorithms through the unifying principle of *structural alignment*, where the dimensionality and nature of the surrogate prior matches the operational mechanics of the black-box search algorithm. This principle guides our methodological decisions, ensuring that distinct types of transferable priors naturally align with specific optimization paradigms. More specifically, we focus on two transferable properties that provide information about the attack surface:

**Decision Boundary Geometry.** Models trained on similar data distributions learn correlated decision boundaries. Adversarial examples lying near a surrogate's decision boundary are statistically enriched near the target's boundary (Section 4.2.4). This geometric correlation provides *spatial information* indicating which regions of input space are promising for adversarial search.

**Loss Landscape Topography.** Gradient-based saliency maps reveal input regions where the loss function is most sensitive to perturbations. Empirical evidence shows that these sensitivity patterns transfer across architecturally similar models, even when exact gradient vectors differ. This topological information provides *directional guidance*, indicating which perturbations are most likely to increase loss.

We focus on these properties because they represent fundamentally different aspects of transferability that is directly exploited in query-based attacks. Boundary geometry provides information about the global structure of the adversarial space where misclassification regions exist, which is critical for initializing search algorithms effectively. Loss topography provides information about the local sensitivity landscape, which input regions are most vulnerable to perturbation, which is essential for guiding iterative refinement. Together,

they span the spectrum from global positioning to local gradient information, enabling complementary exploitation strategies for query-efficient attacks.

We incorporate these properties into SOTA query-based attacks through two distinct approaches: TGEA (Section 4.2.4) leverages boundary geometry to warm-start evolutionary optimization with surrogate-evolved populations, achieving up to 98% attack success rates while reducing queries by 30–60% compared to random initialization. SGSA (Section 4.3.3) leverages loss topography to guide Square Attack's random perturbations toward salient regions, achieving query reductions of 30–50% while maintaining competitive success rates. TGEA and SGSA are operationally independent algorithms, not sequential steps in a pipeline. However, they are strategically complementary, targeting different transferable properties at different search scales.

Ultimately, these methods demonstrate that understanding *what* transfers enables principled design of *how* to exploit it. By exploring two distinct types of transferable information, we demonstrate that effective exploitation depends on matching the information structure to the search algorithm's operational characteristics. Thus, this framework not only achieves substantial query reductions across diverse threat models, but also provides a foundation for future work to identify and exploit additional transferable properties in black-box adversarial settings.

## 4.2 Transfer-Guided Evolutionary Attack (TGEA)

### 4.2.1 Overview

Traditional evolutionary attacks on black-box models have been widely used in the literature (Qiu et al., 2021; Ilyas et al., 2018; Alzantot et al., 2019). They rely on randomly initialized populations, which we empirically demonstrate can lead to relatively slower convergence and higher query costs. This is particularly problematic in scenarios where query budgets are limited or the decision boundaries of the target model are complex. To address these limitations, we introduce the **Transfer-Guided Evolutionary Attack (TGEA)**, a framework that leverages information from a surrogate model to initialize the population of an evolutionary algorithm, which acts as a powerful global search method. By using transfer-based priors, we demonstrate that TGEA improves the efficiency of the attack process and significantly reduces query complexity.

The core idea of TGEA is to build upon the hybrid attack paradigm by bridging the gap between transfer learning and evolutionary optimization. As established by (Suya et al., 2020), this approach uses high-quality adversarial examples generated on a surrogate model to provide a "warm start" for the attack on the black-box model. However, while (Suya et al., 2020) typically transfers a *single* adversarial candidate to initialize a local gradient estimator (e.g., NES or AutoZOOM), TASI and SEGI utilize surrogates to construct a diverse *population* of candidates. By estimating both the mean and the covariance of these candidates, our method warm-starts the CMA-ES search distribution rather than just a single trajectory. This captures the structural geometry of the adversarial subspace, effectively defining a directed **search cone** rather than a single point; this population-based alignment smooths out local irregularities, allowing the optimizer to step over non-transferable traps and converge toward the global optimum.

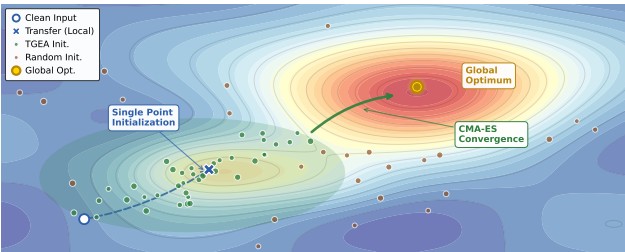

Figure 1: Schematic illustration of TGEA's initialization strategy. TGEA's search cone (green) uses surrogate evolved populations to initialize CMA-ES, enabling convergence to the global optimum (gold), unlike single-point transfer attacks (blue) that get trapped in local optima or random initialization (brown) that lacks guidance.

### 4.2.2 Population Initialization Strategies

**Transfer-Attack Seeded Initialization (TASI).** TASI generates diverse initial candidates by applying multiple types of attacks to the surrogate model, effectively serving as a mechanism to combine multiple powerful attack strategies into a single, unified initialization pool. For a clean input $x$, we construct a diverse set of $k$ transfer attacks $\mathcal{A} = \{A_1, A_2, \ldots, A_k\}$ (e.g., PGD (Madry et al., 2019), Square Attack (Andriushchenko et al., 2020), Boundary Attack (Brendel et al., 2018)).

For each attack $A_i$, we generate a base adversarial example $x_i^{\mathrm{adv}} = A_i(x, f_s)$ and create $q$ local variants by adding Gaussian noise:

$$x_{i,j} = x_i^{\mathrm{adv}} + \mathcal{N}(0, \sigma^2 I), \quad j = 1, \ldots, q \tag{8}$$

The final population is $P_0 = \{x_{i,j}\}_{i=1,\ldots,k;j=1,\ldots,q}$ with $|P_0| = N$.

**Surrogate-Evolved Genetic Initialization (SEGI).** SEGI employs a meta-optimization process, running a Genetic Algorithm (GA) entirely on the surrogate model to evolve a high-quality initial population. Starting from a random population of size $N$, we evolve it for $G$ generations using a surrogate-specific fitness function:

$$\mathcal{F}_s(\delta) = \alpha_s \cdot \max_{j \neq y} f_{s,j}(x + \delta) - \gamma_s \cdot f_{s,y}(x + \delta) + \delta \cdot D_{\mathrm{centroid}}(\delta) + \mathcal{R}_{\mathrm{div}}(P) \tag{9}$$

where $f_{s,j}$ denotes the $j$-th class score from surrogate $f_s$, and $\mathcal{R}_{\mathrm{div}}$ is a diversity penalty:

$$\mathcal{R}_{\mathrm{div}}(P) = -\epsilon \cdot \frac{1}{|P|} \sum_{p \in P} \mathrm{MSE}(\delta, p) \tag{10}$$

The term $D_{\mathrm{centroid}}(\delta)$ guides the search in the latent space:

$$D_{\mathrm{centroid}}(\delta) = \|z(x + \delta) - c_y\|_2 - \min_{i \neq y} \|z(x + \delta) - c_i\|_2 \tag{11}$$

where $z(\cdot)$ is the penultimate layer embedding and $c_i$ is the centroid of class $i$ in the latent space (computed on the surrogate). This encourages the GA to maintain population diversity and steer candidates toward incorrect class clusters, preventing premature convergence.

After $G$ generations, we select the top $N$ individuals by fitness, add small Gaussian noise to each, and use them to initialize CMA-ES on the target model.

Our fitness functions build upon the frameworks established in (Huang & Yu, 2022) and (Qiu et al., 2021), but extend them by assigning adjustable weights to each component term. These weights were empirically calibrated through iterative experimentation to optimize overall performance. It is important however to note that unlike (Huang & Yu, 2022) which uses standard genetic operators, our TGEA employs CMA-ES optimization.

### 4.2.3 CMA-ES Refinement

Given the initialized population $P_0$, we apply CMA-ES to optimize the target model's loss. The fitness function for the target model is:

$$\mathcal{F}_t(\delta) = \alpha_t \cdot \max_{j \neq y} f_{t,j}(x + \delta) - \gamma_t \cdot f_{t,y}(x + \delta) \tag{12}$$

We deliberately distinguish the weighting parameters $(\alpha_t, \gamma_t)$ from those used in the surrogate stage $(\alpha_s, \gamma_s)$ to prioritize exploitation of the target model's specific decision boundary over the exploration-heavy objective used during initialization. View Appendix B1 for ablations. CMA-ES iteratively updates the mean vector $m^{(t)}$, step size $\sigma^{(t)}$, and covariance matrix $C^{(t)}$ as described in (Hansen, 2006).

### 4.2.4 Design Rationale and Intuitive Motivation

**Decision Boundary Transferability.** TGEA leverages the empirically observed phenomenon that models trained on similar data learn correlated decision boundaries (Papernot et al., 2017; Liu et al., 2017). We define the near-boundary region with margin threshold $\tau$ as:

$$\mathcal{B}_f^\tau = \{x : |\text{margin}(f(x))| < \tau\} \tag{13}$$

where $\text{margin}(f(x)) = f_y(x) - \max_{j \neq y} f_j(x)$.

Prior work suggests that adversarial examples near the decision boundary of a surrogate model are often also near the boundary of similar target models (Tramèr et al., 2017). While the exact enrichment factor varies with architecture similarity, we qualitatively expect:

$$\mathbb{P}(x \in \mathcal{B}_t^\epsilon \mid x \in \mathcal{B}_s^\epsilon) > \mathbb{P}(x \in \mathcal{B}_t^\epsilon) \tag{14}$$

**Expected Query Reduction.** By initializing CMA-ES with examples from $\mathcal{B}_s^\epsilon$, we essentially hypothesize that the search starts closer to $\mathcal{B}_t^\epsilon$ compared to random initialization. Since CMA-ES convergence depends on the initial distance to the optimum (Hansen & Ostermeier, 2001), transfer-based initialization should reduce the number of queries required.

To validate this hypothesis, we compare queries required by TGEA versus random initialization across our experiments. Our results show that TGEA consistently requires fewer queries than random CMA-ES (Tables 1-4), with reductions ranging from 30-60% depending on the dataset and model architecture.

## 4.3 Saliency-Guided Square Attack (SGSA)

### 4.3.1 Motivation

The Square Attack (Andriushchenko et al., 2020) achieves state of the art query efficiency through random square-shaped perturbations that align with CNN inductive biases, specifically exploiting the local spatial correlations learned by convolutional architectures. By perturbing contiguous square regions rather than individual pixels, Square Attack naturally targets the receptive field structures that CNNs are sensitive to, making it highly effective in black-box settings. However, despite this architectural alignment, its purely random search strategy treats all spatial locations uniformly, allocating queries without regard to their potential impact on model predictions. This leads to significant query waste on irrelevant or low-sensitivity image regions such as uniform backgrounds, occluded areas, or regions far from decision boundaries that contribute minimally to inducing misclassification.

We empirically demonstrate that this limitation can be addressed by incorporating surrogate-derived saliency maps to provide spatial guidance for the attack. Specifically, gradient-based saliency maps computed on an accessible surrogate model identify regions where the loss function exhibits high sensitivity to input perturbations. By leveraging the empirical observation that such high-gradient regions often transfer across architecturally similar models, we target Square Attack's random search toward sensitive areas. This approach preserves Square Attack's strengths while intelligently focusing computational resources on perturbations most likely to succeed, thereby further reducing query costs without sacrificing attack success rates.

### 4.3.2 Hybrid Guidance Map Generation

SGSA constructs a dynamic vulnerability heatmap $S(x)$ by combining multi-scale saliency and attention mechanisms from the surrogate model.

**Multi-Scale Saliency.** We compute the gradient magnitude of the surrogate loss with respect to the input at multiple scales $\mathcal{S} = \{s_1, s_2, s_3\}$:

$$S_{\text{grad}}^{(s)}(x) = \|\nabla_x L(f_s(x), y)\|_2 \tag{15}$$

$$S_{\text{multi}}(x) = \frac{1}{|\mathcal{S}|} \sum_{s \in \mathcal{S}} \text{Upsample}(\text{Smooth}(S_{\text{grad}}^{(s)}(x))) \tag{16}$$

The multi-scale approach captures different levels of feature abstraction. By incorporating downsampled scales, we effectively apply a low-pass filter to the gradient information, guiding the attack towards structural, low frequency vulnerabilities that are more transferable across architectures than high frequency noise (Ilyas et al., 2018). Specifically, high frequency gradient features (pixel level noise, texture artifacts) are model specific, while low frequency structural patterns (object boundaries, semantic region importance) represent universal vulnerability patterns shared across architectures. Downsampling suppresses model specific noise while retaining transferable structural signals; subsequent upsampling restores spatial resolution for precise perturbation placement.

**Attention Map Integration.** We optionally incorporate an attention mechanism $A(x)$ from the surrogate:

$$S_{\text{hybrid}}(x) = (1 - \beta) \cdot S_{\text{multi}}(x) + \beta \cdot A(x) \tag{17}$$

where $\beta = 0.3$ balances gradient sensitivity and model attention.

**Temporal Smoothing.** As the adversarial example evolves, we update $S_{\text{hybrid}}$ periodically and apply exponential moving average:

$$S^{(t+1)} = \lambda \cdot S^{(t)} + (1 - \lambda) \cdot S_{\text{hybrid}}(x^{(t)}) \tag{18}$$

with $\lambda = 0.9$ to maintain stability while adapting to perturbation changes.

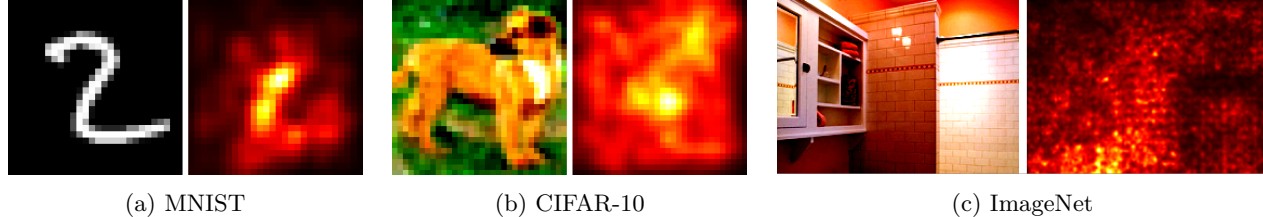

| (a) MNIST | (b) CIFAR-10 | (c) ImageNet |

Figure 2: Examples of saliency maps generated from surrogate models (left: original, right: map). Brighter regions indicate higher model sensitivity and thus represent more promising areas to attack.

### 4.3.3 Guided Perturbation Placement and Sizing

At iteration $t$, SGSA selects the square location and size based on $S^{(t)}$.

**Probabilistic Location Sampling.** We first normalize $S^{(t)}$ to create a valid probability mass function over pixel locations:

$$\pi_{i,j}^{(t)} = \frac{S_{i,j}^{(t)}}{\sum_{u,v} S_{u,v}^{(t)}} \tag{19}$$

We then sample the top-left corner $(r, c)$ of the square from this distribution:

$$(r, c) \sim \text{Categorical}(\pi^{(t)}) \tag{20}$$

This focuses perturbations on high-saliency regions.

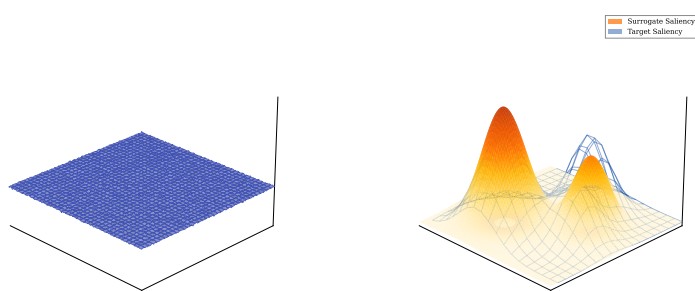

Figure 3: Comparison of sampling strategies for black-box adversarial attacks. **Left:** Uniform sampling allocates equal probability across all spatial locations via a uniform distribution, characteristic of random search methods like Square Attack. **Right:** Saliency-guided sampling uses a probability distribution concentrated on high-gradient regions identified through surrogate model gradients (warm surface) that correlate with target model vulnerabilities (blue wireframe), enabling more efficient perturbation placement.

**Adaptive Size Adjustment.** The square side length $h^{(t)}$ is computed via an inverse relationship with local saliency:

$$h^{(t)} = h^{(t)}_{\text{base}} \cdot \text{saliency\_factor}(r, c) \tag{21}$$

where:

$$\text{saliency\_factor}(r, c) = \frac{1}{1 + \alpha_{\text{scale}} \cdot S^{(t)}_{r,c}} \tag{22}$$

High saliency leads to smaller, precise perturbations; low saliency to larger, exploratory ones.

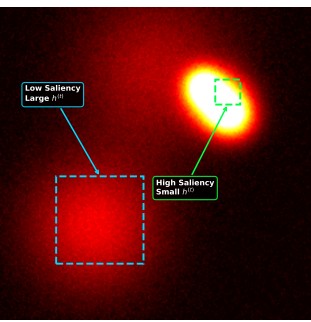

Figure 4: SGSA adaptive sizing mechanism. The saliency map shows gradient magnitude from the surrogate model. Large perturbation squares (cyan, dashed) are placed in low-saliency regions for exploration, while small squares (green, dashed) target high-saliency regions for precise attacks.

### 4.3.4 Fallback Mechanism

To ensure robustness against poor transferability, SGSA includes dual fallback triggers:

- **Stagnation-based:** If loss does not improve for $T_{\text{stag}}$ iterations, revert to random search for the next $T_{\text{rand}}$ iterations.

- **Stochastic:** With probability $p_{\text{rand}}$, perform a random square placement to escape local optima.

### 4.3.5 Design Rationale and Empirical Validation

**Motivation from White-Box Attacks.** The core principle of SGSA, prioritizing perturbations in high-gradient regions, mirrors the fundamental mechanism of gradient-based white-box attacks. Methods like PGD and FGSM explicitly compute:

$$\delta = \epsilon \cdot \text{sign}(\nabla_x L(f(x), y)) \tag{23}$$

which concentrates perturbation magnitude where gradients are largest, i.e., where the loss is most sensitive to input changes. SGSA extends this principle to the black-box setting by using surrogate gradients to guide random search.

**Local Gradient Transferability Hypothesis.** SGSA leverages multi-scale saliency patterns from surrogate models to guide perturbation placement. Let $S_s(x)_i = \|\nabla_{x_i} L(f_s(x), y)\|_2$ and $S_t(x)_i = \|\nabla_{x_i} L(f_t(x), y)\|_2$ denote gradient magnitude saliency maps for surrogate and target models, respectively.

We measured the Spearman rank correlation between surrogate (ResNet18) and target model multi-scale saliency maps on 1,000 correctly classified CIFAR-10 samples:

**Measured Correlations:**

- CIFAR-10 standard models: $\rho = 0.68 \pm 0.16$ (InceptionV3), $\rho = 0.65 \pm 0.18$ (VGG16)

- CIFAR-10 robust models: $\rho = 0.41 \pm 0.20$ (WideResNet), $\rho = 0.39 \pm 0.19$ (WRN-94-16)

Multi-scale averaging yields 55-63% improvement over single-scale for standard models and 160% for robust models (all $p < 0.001$). The moderate correlations ($\rho = 0.39\text{-}0.68$) demonstrate that saliency patterns transfer sufficiently to guide black-box attacks, with standard models showing stronger correlation than adversarially trained models due to their distinct loss landscapes (Qin et al., 2019).

**Query Efficiency Through Guided Search.** SGSA prioritizes perturbations in high-saliency regions, which we expect to be more effective for causing misclassification. To understand the potential query savings, consider a simplified model where:

- High-saliency regions $\mathcal{H}$ comprise fraction $\alpha$ of the input space

- Due to correlation $\rho$, SGSA places perturbations in effective regions with higher probability

- Perturbations in $\mathcal{H}$ are $\lambda$ times more effective than in low-saliency regions $\mathcal{L}$

Under these assumptions, we can heuristically approximate the expected query reduction as:

$$\mathbb{E}[Q_{\text{SGSA}}] \approx \frac{1}{1 + \rho\alpha(\lambda - 1)} \cdot \mathbb{E}[Q_{\text{Square}}] \tag{24}$$

While this is an oversimplification (it assumes binary effectiveness, independence, and perfect guidance), it provides intuition for why guidance helps. For typical conservative values, $\alpha = 0.2$ (CNN saliency concentrates on roughly 20% of spatial locations corresponding to object boundaries), $\rho = 0.65$ (our measured surrogate target correlations from Section 4.3.5), and $\lambda = 3$ (reflecting the expectation that perturbations in high saliency regions are significantly more effective at inducing misclassification than changes to the background) the model predicts approximately 21% query reduction. While our empirical results show greater reductions (30-50% in Tables 4 and 5), this simplified model provides intuition for the fundamental mechanism: by concentrating search on regions where surrogate gradients transfer, SGSA reduces the expected queries needed to find successful perturbations.

**Adaptive Sizing Rationale.** Our adaptive sizing rule $h^{(t)} = h_{\text{base}}^{(t)}/(1 + \alpha_{\text{scale}} \cdot S_{r,c}^{(t)})$ is motivated by the intuition that:

- In high-saliency regions (steep local loss), smaller perturbations provide more precise control

- In low-saliency regions (flatter loss), larger perturbations enable faster exploration

# 5 Experimental Evaluation

## 5.1 Experimental Setup

### 5.1.1 Datasets and Models

We evaluate our framework on three datasets of increasing complexity to test scalability and robustness:

- **MNIST:** 10,000 grayscale $28 \times 28$ handwritten digit images. We target two models: Model A (CNN architecture from (Tramèr et al., 2020)) and Model B (architecture from (Carlini & Wagner, 2017)), both trained in-house. The surrogate is a lightweight 2-layer CNN.

- **CIFAR-10:** 10,000 RGB $32 \times 32$ images across 10 classes. We evaluate on standard pretrained models: InceptionV3 (Szegedy et al., 2016) and VGG16 (Simonyan & Zisserman, 2015). To assess performance against defenses, we test on adversarially trained models: WideResNet (Carmon et al., 2019) and WRN-94-16 (Bartoldson et al., 2024), the latter being the current state-of-the-art on RobustBench (Croce et al., 2021). The surrogate is ResNet18 (He et al., 2016). Non robust model weights used are from (Phan, 2021).

- **ImageNet:** 1,000 validation images from ImageNet-mini (Figotin, 2019) ($224 \times 224$ RGB). Targets include DenseNet-121 (Huang et al., 2018) and EfficientNet-B1 (Tan & Le, 2019). The surrogate is ResNet18.

### 5.1.2 Attack Configuration

We benchmark PriSM against state-of-the-art methods categorized in Table 1. All attacks operate under the $L_\infty$ norm. We set perturbation budgets $\epsilon \in \{0.2, 0.3\}$ for MNIST, $\{0.1, 0.2\}$ for CIFAR-10, and $\{0.05, 0.1\}$ for ImageNet. The maximum query budget is fixed at $Q_{\max} = 1000$. We sample $\sim$1000 correctly classified images per dataset.

Table 1: Summary of evaluated black-box attack methods.

| Method | Type | Description |
|---|---|---|
| **TGEA-TASI (Ours)** | Hybrid | Transfer attacks + CMA-ES |
| **TGEA-SEGI (Ours)** | Hybrid | GA-evolved surrogate + CMA-ES |
| **SGSA (Ours)** | Hybrid | Saliency-Guided Square Attack |
| Random CMA-ES | Score-based | Randomly initialized CMA-ES |
| Square Attack (Andriushchenko et al., 2020) | Score-based | Random square perturbations |
| SimBA (Guo et al., 2019) | Score-based | Coordinate-wise random search |
| HSJA (Chen et al., 2020) | Decision-based | HopSkipJumpAttack boundary search |
| PBO (Cheng et al., 2024) | Hybrid | Bayesian optimization with function prior |

## 5.2 Results

### 5.2.1 MNIST

Tables 2 and 3 summarize performance on MNIST. The simple dataset structure favors local search strategies. **SGSA** achieves the highest Attack Success Rate (ASR) of 99.81–100% while maintaining low query costs (54.37–181.80 queries). PBO demonstrates exceptional query efficiency (as low as 14.15 queries) but occasionally trails SGSA in success rate on Model A. TGEA variants show competitive ASR but higher query costs, as global optimization is less critical for this lower-dimensional manifold.

Table 2: ASR and AQ on MNIST ($\ell_\infty$, $\epsilon = 0.2$).

| | Model A | | Model B | |
|---|---|---|---|---|
| Method | ASR ↑ | AQ ↓ | ASR ↑ | AQ ↓ |
| **TGEA-TASI** | 65.76 | 250.05 | 82.11 | 197.74 |
| **TGEA-SEGI** | 47.15 | 255.86 | 60.34 | 272.78 |
| Random | 37.77 | 519.96 | 51.37 | 556.02 |
| Square | 87.86 | 239.38 | 91.79 | 264.05 |
| **SGSA** | **89.89** | 177.48 | **93.98** | 181.80 |
| SimBA | 59.32 | 306.05 | 69.07 | 265.60 |
| HSJA | 20.23 | 431.29 | 26.48 | 483.21 |
| PBO | 70.39 | **25.36** | 89.05 | **14.15** |

Table 3: ASR and AQ on MNIST ($\ell_\infty$, $\epsilon = 0.3$).

| | Model A | | Model B | |
|---|---|---|---|---|
| Method | ASR ↑ | AQ ↓ | ASR ↑ | AQ ↓ |
| **TGEA-TASI** | 91.76 | 130.24 | 97.13 | 74.94 |
| **TGEA-SEGI** | 83.69 | 164.57 | 90.78 | 125.23 |
| Random | 76.85 | 398.30 | 88.39 | 357.62 |
| Square | **99.93** | 91.45 | 100.00 | 97.73 |
| **SGSA** | 99.81 | **54.37** | **100.00** | **56.68** |
| SimBA | 94.14 | 199.98 | 81.50 | 151.06 |
| HSJA | 72.09 | 407.43 | 76.04 | 398.73 |
| PBO | 97.44 | 99.75 | 99.70 | 67.30 |

### 5.2.2 CIFAR-10

Results for standard CIFAR-10 models are presented in Tables 4 and 5. A strategic trade off emerges: **TGEA-SEGI** achieves the highest ASR (up to 98.36%) among evolutionary methods, proving effective for maximizing attack success. **PBO** excels in query efficiency, achieving the lowest AQ with high ASR.

Table 4: ASR and AQ on CIFAR-10 ($\ell_\infty$, $\epsilon = 0.1$).

| | InceptionV3 | | VGG16 | |
|---|---|---|---|---|
| Method | ASR ↑ | AQ ↓ | ASR ↑ | AQ ↓ |
| **TGEA-TASI** | 89.69 | 120.75 | 88.03 | 180.28 |
| **TGEA-SEGI** | **91.43** | 105.26 | **89.80** | 127.82 |
| Random | 85.27 | 148.62 | 83.29 | 198.42 |
| Square | 88.20 | 76.64 | 81.56 | 91.70 |
| **SGSA** | 85.64 | 53.47 | 82.85 | 77.62 |
| SimBA | 75.89 | 105.63 | 76.33 | 163.42 |
| HSJA | 56.39 | 368.84 | 40.07 | 442.09 |
| PBO | 88.69 | **52.86** | 87.80 | **47.95** |

Table 5: ASR and AQ on CIFAR-10 ($\ell_\infty$, $\epsilon = 0.2$).

| | InceptionV3 | | VGG16 | |
|---|---|---|---|---|
| Method | ASR ↑ | AQ ↓ | ASR ↑ | AQ ↓ |
| **TGEA-TASI** | 97.10 | 53.82 | 97.20 | 59.73 |
| **TGEA-SEGI** | 98.36 | 27.95 | 98.19 | 34.80 |
| Random | 95.56 | 78.30 | 92.61 | 80.85 |
| Square | 94.39 | 38.15 | 91.33 | 41.95 |
| **SGSA** | 93.04 | 30.21 | 91.04 | 34.05 |
| SimBA | 77.92 | 81.80 | 77.94 | 104.60 |
| HSJA | 92.67 | 240.01 | 77.71 | 255.88 |
| PBO | **98.67** | **16.87** | **99.43** | **19.33** |

**Robust Models.** Tables 6 and 7 evaluate attacks on adversarially trained models. Typically, these models mask gradients, making attacks harder. However, we observe a distinct phenomenon where SGSA outperforms baselines significantly. We hypothesize it is because adversarial training creates "perceptually aligned" gradients, meaning the gradient of a robust surrogate is more interpretable and structurally correlated with the target model than that of a non robust model (Tsipras et al., 2019). SGSA possibly leverages this restored correlation to break the traditional efficiency-success trade off, achieving the highest ASR (61.24%) on the SOTA robust model WRN-94-16.

Table 6: ASR and AQ on robust models ($\ell_\infty$, $\epsilon = 0.1$).

| | WideResNet | | WRN-94-16 | |
|---|---|---|---|---|
| Method | ASR ↑ | AQ ↓ | ASR ↑ | AQ ↓ |
| **TGEA-TASI** | 40.23 | 135.09 | 32.17 | 124.62 |
| **TGEA-SEGI** | 37.08 | 140.98 | 31.27 | 178.06 |
| Random | 38.11 | 143.05 | 31.58 | 189.83 |
| Square | 42.09 | 115.60 | 32.92 | 174.91 |
| **SGSA** | **49.38** | 52.78 | **37.99** | 120.44 |
| SimBA | 41.95 | 82.14 | 30.12 | 195.92 |
| HSJA | 38.62 | 77.46 | 26.32 | **73.93** |
| PBO | 45.52 | **22.03** | 31.58 | 124.72 |

Table 7: ASR and AQ on robust models ($\ell_\infty$, $\epsilon = 0.2$).

| | WideResNet | | WRN-94-16 | |
|---|---|---|---|---|
| Method | ASR ↑ | AQ ↓ | ASR ↑ | AQ ↓ |
| **TGEA-TASI** | 60.82 | 181.41 | 55.71 | 225.07 |
| **TGEA-SEGI** | 59.32 | 178.22 | 52.33 | 225.51 |
| Random | 60.52 | 178.84 | 54.65 | 254.03 |
| Square | 65.07 | 116.10 | 60.65 | 151.02 |
| **SGSA** | 68.20 | **67.94** | 61.24 | 123.70 |
| SimBA | 51.15 | 125.47 | 46.20 | 293.47 |
| HSJA | 48.02 | 123.61 | 29.19 | **54.89** |
| PBO | **69.16** | 92.25 | **62.11** | 141.93 |

### 5.2.3 ImageNet

Tables 8 and 9 summarize performance on ImageNet models. SGSA achieves superior query efficiency on DenseNet (55.85-75.91 queries) while TGEA-SEGI maximizes ASR on EfficientNet (76.88-89.10%), demonstrating scalability to high dimensional inputs.

Table 8: ASR and AQ on ImageNet ($\ell_\infty$, $\epsilon = 0.05$).

|  | DenseNet | | EfficientNet | |
| --- | --- | --- | --- | --- |
| Method | ASR ↑ | AQ ↓ | ASR ↑ | AQ ↓ |
| **TGEA-TASI** | 75.43 | 99.05 | 72.73 | 188.89 |
| **TGEA-SEGI** | 72.50 | 100.08 | **76.88** | 193.62 |
| Random | 72.50 | 98.79 | 76.88 | 182.77 |
| Square | 82.01 | 116.57 | 69.16 | 137.35 |
| **SGSA** | **85.59** | **75.91** | 72.87 | 102.31 |
| SimBA | 66.77 | 53.04 | 50.55 | **64.33** |
| HSJA | 42.86 | 77.33 | 18.10 | 116.00 |
| PBO | 71.43 | 116.60 | 61.21 | 167.55 |

Table 9: ASR and AQ on ImageNet ($\ell_\infty$, $\epsilon = 0.1$).

|  | DenseNet | | EfficientNet | |
| --- | --- | --- | --- | --- |
| Method | ASR ↑ | AQ ↓ | ASR ↑ | AQ ↓ |
| **TGEA-TASI** | 88.63 | 128.01 | 83.82 | 158.75 |
| **TGEA-SEGI** | 89.47 | 111.25 | **89.10** | 148.42 |
| Random | 89.10 | 110.78 | 86.84 | 130.38 |
| Square | 91.84 | 79.93 | 84.71 | 118.79 |
| **SGSA** | **93.99** | **55.85** | 86.99 | **89.56** |
| SimBA | 71.47 | 79.29 | 55.06 | 97.45 |
| HSJA | 30.16 | 98.23 | 28.65 | 171.39 |
| PBO | 93.33 | 100.67 | 81.87 | 167.02 |

To assess the generalization of our methods beyond convolutional neural networks, we extend our evaluation to newer, Vision Transformer architectures. We perform experiments on two representative models: the standard Vision Transformer (ViT-B/16) (Dosovitskiy et al., 2021) and the Data-efficient Image Transformer (DeiT-S) (Touvron et al., 2021). These experiments demonstrate that priors derived from CNN surrogates (spatial saliency and decision boundary geometry) can effectively transfer to the distinct loss landscapes of transformer-based models.

Table 10: ASR and AQ on ViT/DeiT ($\ell_\infty$, $\epsilon = 0.05$).

|  | ViT-B/16 | | DeiT-S | |
| --- | --- | --- | --- | --- |
| Method | ASR ↑ | AQ ↓ | ASR ↑ | AQ ↓ |
| **TGEA-TASI** | **64.26** | 215.30 | 61.36 | 202.65 |
| **TGEA-SEGI** | 55.05 | 241.17 | 57.33 | 275.49 |
| Random | 53.42 | 248.62 | 56.67 | 267.22 |
| Square | 56.93 | 131.16 | 62.99 | 132.11 |
| **SGSA** | 61.59 | **111.10** | **70.10** | **105.14** |
| SimBA | 32.63 | 240.42 | 47.21 | 243.12 |
| HSJA | 24.82 | 180.42 | 42.13 | 201.32 |
| PBO | 53.15 | 193.10 | 63.25 | 191.50 |

Table 11: ASR and AQ on ViT/DeiT ($\ell_\infty$, $\epsilon = 0.1$).

|  | ViT-B/16 | | DeiT-S | |
| --- | --- | --- | --- | --- |
| Method | ASR ↑ | AQ ↓ | ASR ↑ | AQ ↓ |
| **TGEA-TASI** | **77.15** | 193.95 | 79.16 | 197.41 |
| **TGEA-SEGI** | 74.22 | 208.98 | 75.10 | 178.05 |
| Random | 72.11 | 219.93 | 73.43 | 198.04 |
| Square | 69.86 | 139.33 | 82.25 | 124.67 |
| **SGSA** | 75.74 | **110.40** | **87.90** | **104.52** |
| SimBA | 43.24 | 221.42 | 46.21 | 243.12 |
| HSJA | 47.91 | 193.32 | 36.42 | 202.42 |
| PBO | 62.61 | 186.51 | 72.23 | 189.13 |

## 5.3 Sensitivity Analysis and Component Validation

To demonstrate the robustness of our hyperparameters and identify optimal configurations, we conducted a systematic sensitivity analysis for both attacks. We present the key findings here, with comprehensive experimental results and additional ablation details available in **Appendix B**. The results, summarized in Figure 6 and Figure 5, highlight key trade-offs between hyperparameter choices and attack performance.

### 5.3.1 TGEA Configuration and Fitness Terms

Figure 5 analyzes the evolutionary components of TGEA:

- **Impact of Fitness Terms (Figure 5a):** We performed an ablation study on the SEGI fitness function components. The results show that removing the target class penalty $\gamma_s$, unsurprisingly, causes the most severe degradation, particularly for DenseNet (green bar drops significantly). This confirms that explicitly penalizing the true label is critical for evolving effective "search cones" that steer the population away from the original class manifold.

- **Population Size (Figure 5d):** We tested CMA-ES population sizes of $\{6, 16, 26\}$. The data indicates an optimal size of 16 across most models. Smaller populations (6) lack sufficient diversity to escape local optima, while larger populations (26) appear to dilute the search pressure, leading to slower convergence and lower ASR, especially for VGG16 (orange line).

- **Weight Balancing (Figure 5b-c):** The attack shows stability around a target weight $\alpha_t \approx 1.0$. However, increasing the target penalty $\gamma_t$ (Figure 5c) yields significant gains for DenseNet, suggesting that complex models might benefit from more aggressive penalties on the true class probability.

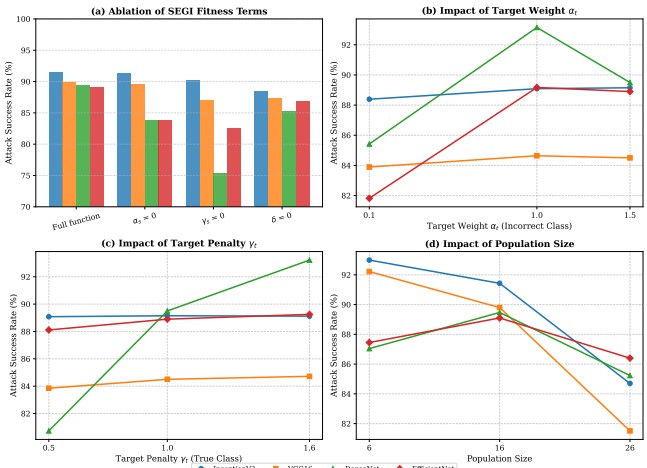

Figure 5: **Sensitivity Analysis of TGEA Configuration.** We evaluate (a) Ablation of SEGI Fitness terms, (b) Target Weight $\alpha_t$, (c) Target Penalty $\gamma_t$, and (d) CMA-ES Population Size. The analysis reveals that the target penalty term is critical for success (a) and that a population size of 16 provides the optimal balance between exploration and exploitation (d).

### 5.3.2 SGSA Hyperparameter Sensitivity

Figure 6 provides a comprehensive breakdown of how SGSA's components influence success rates:

- **Saliency Scale (Figure 6a):** We notice a positive correlation between the scale factor and ASR, particularly for VGG16 and InceptionV3. A factor of 10 consistently outperforms smaller factors (2 or 5), suggesting that stronger Gaussian smoothing is essential for extracting stable, transferable gradients from the surrogate.

- **Attention Weight $\beta$ (Figure 6b):** The attack is relatively robust to $\beta$ values between 0.1 and 0.75, with a peak performance typically around $\beta = 0.3$. Notably, setting $\beta = 0$ (no attention) leads to a sharp drop in ASR for InceptionV3 and VGG16, validating the necessity of the attention mechanism.

- **Update Frequency $k$ (Figure 6c):** The update frequency controls the trade-off between gradient freshness and computational cost. While $k = 50$ offers a stable sweet spot, increasing $k$ to 100 (lazier updates) causes a noticeable performance degradation on complex architectures like EfficientNet (red line), indicating that high dimensional landscapes require more frequent gradient refreshes.

- **Temporal Decay $\lambda$ (Figure 6d):** Higher decay values tend to yield better results, suggesting that retaining a longer history of gradients helps stabilize the search direction against local noise.

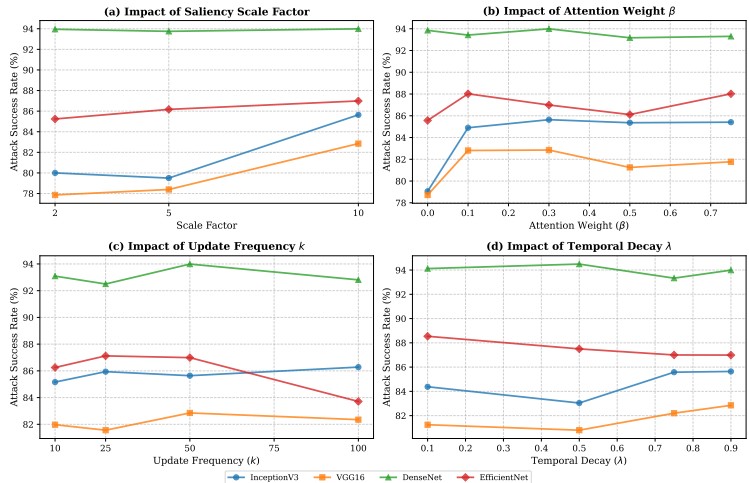

Figure 6: **Sensitivity Analysis of SGSA Hyperparameters.** We analyze the impact of (a) Saliency Scale Factor, (b) Attention Weight $\beta$, (c) Saliency Update Frequency $k$, and (d) Temporal Decay $\lambda$ across four target architectures. The results validate that stronger smoothing, moderate attention, and high temporal retention generally are more robust.

# 6  Analysis and Discussion

Our experimental evaluation across MNIST, CIFAR-10, and ImageNet reveals consistent patterns in the performance characteristics of PriSM's two approaches.

## 6.1  Performance Against Baselines

**Substantial Improvements Over Established Baselines.** Both SGSA and TGEA demonstrate significant improvements over established baseline methods across all experimental settings. SGSA achieves 30–50% query reductions compared to Square Attack while maintaining or exceeding success rates. For instance, on InceptionV3 ($\epsilon = 0.1$), SGSA requires only 53.47 queries versus 76.64 for Square Attack, a 30.23% reduction, while maintaining competitive ASR (85.64% vs 88.20%). Similarly, TGEA consistently outperforms Random CMA-ES initialization by 30–60% in query efficiency, validating the fundamental value of surrogate-guided initialization for evolutionary methods. On CIFAR-10 at $\epsilon = 0.2$, TGEA-SEGI requires only 27.95 queries compared to Random CMA-ES's 78.30 queries while achieving higher success rates (98.36% vs 95.56%).

Against other gradient free baselines like SimBA and HSJA, PriSM methods show even more pronounced advantages. On ImageNet models, SGSA reduces queries by 40–50% compared to SimBA while achieving 15–20% higher success rates. These systematic improvements across diverse baselines confirm that exploiting transferable surrogate provides substantial benefits over uninformed search strategies.

**Competitive Performance with SOTA PBO.** PBO (Cheng et al., 2024) represents the current state of the art in query-efficient black-box attacks, leveraging Bayesian optimization with learned function priors. Our evaluation reveals that PriSM achieves competitive or superior performance depending on the attack scenario, with clear complementary strengths.

On CIFAR-10 standard models, PBO achieves exceptional query efficiency (16.87 vs 52.86 queries) with high success rates, outperforming both SGSA and TGEA variants in this regime. This advantage stems from Bayesian optimization's sophisticated modeling of loss surfaces through Gaussian processes, which is particularly effective on simpler, well-behaved optimization landscapes where the GP function prior can accurately approximate the target model's loss function.

However, on adversarially trained models, the performance dynamics shift substantially. On WRN-94-16 ($\epsilon = 0.2$), SGSA matches PBO's success rate (61.24% vs 62.11%) while requiring 12.84% fewer queries (123.70 vs 141.93). This suggests that Bayesian optimization's GP-based function prior becomes less effective on the smoothed but highly non convex landscapes of robust models, where SGSA's multi-scale saliency guidance provides more reliable directional information. The measured saliency correlations ($\rho = 0.39$–$0.41$ on robust models) indicate that gradient magnitude patterns transfer sufficiently to guide efficient attacks even when exact loss surfaces diverge.

Additionally, on high resolution ImageNet models, SGSA demonstrates superior efficiency: on DenseNet ($\epsilon = 0.1$), SGSA requires 55.85 queries versus PBO's 100.67, a 44.52% reduction, while achieving comparable success rates (93.99% vs 93.33%). This performance gap highlights a fundamental scalability advantage: saliency-based guidance directly exploits spatial locality in high dimensional image spaces, whereas GP-based function approximation faces increasing complexity with dimensionality. The computational overhead of Bayesian optimization (kernel matrix inversions, acquisition function optimization) grows substantially in higher dimensions, while SGSA's gradient-map computation scales linearly with input size.

## 6.2 SEGI vs. TASI: Quality of Initial Candidates

Our results consistently show TGEA-SEGI achieving higher attack success rates than TGEA-TASI across most experimental settings. This performance gap stems from fundamental differences in how the two methods generate initial candidate populations. TASI constructs its population by combining multiple predefined attack strategies (PGD, Square Attack, Boundary Attack) with Gaussian noise perturbations, essentially performing a diverse but undirected sampling around known attack vectors. In contrast, SEGI employs a meta-optimization process that runs a complete Genetic Algorithm on the surrogate model for $G$ generations, evolving candidates specifically toward the adversarial objective through iterative selection, crossover, and mutation. This surrogate-based evolution allows SEGI to explore the adversarial subspace more systematically, producing a population that has already undergone fitness based refinement before being transferred to the target model. Critically, SEGI's fitness function incorporates both a diversity penalty term $R_{\text{div}}$ (Equation 9) and a centroid distance term $D_{\text{centroid}}$ that encourages movement away from the true class representation in latent space while approaching incorrect class centroids. These components ensure the evolved population maintains coverage of promising regions rather than converging prematurely, providing CMA-ES with a well-distributed initialization that captures the geometric structure of the adversarial manifold while being positioned near decision boundaries. While TASI benefits from the complementary strengths of diverse attack types, SEGI's evolved candidates represent genuinely optimized starting points that have been tailored to the surrogate's loss landscape, which, given the measured transferability correlations, provides better initialization for attacking architecturally similar target models.

## 6.3 Model Architecture and Attack Performance

Our ImageNet experiments (Table 5) reveal architecture-dependent performance patterns. SGSA achieves its strongest results on DenseNet (93.99% ASR, 55.85 queries at $\epsilon = 0.1$), while TGEA-SEGI performs best on EfficientNet (89.10% ASR at $\epsilon = 0.05$).

We observe that models with stronger feature locality (DenseNet's dense connections) tend to exhibit higher saliency transferability, benefiting local search methods like SGSA. Models with more distributed representations (EfficientNet's compound scaling) require the global exploration capabilities of TGEA-SEGI to consistently find adversarial examples. However, we emphasize that these are empirical observations rather than definitive causal relationships; a complete understanding would require systematic landscape analysis beyond the scope of this work.

## 6.4 Practitioner Guidance

Based on our experimental results, we provide the following evidence-based guidance for practitioners. For applications with a **strict query budget**, we recommend **SGSA**, which averaged fewer queries than baseline methods. However, for scenarios where **high attack success rate** (ASR) is the priority, **TGEA-SEGI**

is the better choice, achieving higher ASR (e.g., 98.36% on InceptionV3). In the context of **adversarial training**, **SGSA** demonstrated superior robustness against defenses (e.g., on WRN-94-16). Finally, for **simple datasets** (e.g., MNIST), both methods are highly effective, with SGSA offering a slight advantage in both query efficiency and ASR.

## 7 Limitations and Future Work

While PriSM demonstrates strong empirical performance, we do not provide formal theoretical guarantees on query complexity or convergence rates. Our design rationale (Sections 4.2.4 and 4.3.5) offers intuitive justification based on transferability assumptions, but rigorous analysis would strengthen the theoretical foundation. Nevertheless, our empirical results across diverse settings provides substantial evidence for the practical effectiveness of our approach.

Future work could explore: (1) adaptive surrogate selection strategies that estimate transferability online, (2) meta-learning approaches to automatically tune hyperparameters per target model, (3) theoretical characterization of when saliency-guided local search outperforms global optimization, and (4) extension to other attack settings (e.g., targeted attacks, different threat models).

## 8 Conclusion

We introduced PriSM, a framework that leverages surrogate decision boundary geometry and loss landscape topography to guide black-box attacks. Through TGEA and SGSA, we demonstrated that aligning search strategies with specific surrogate priors reduces query costs by 30–60% across MNIST, CIFAR-10, and ImageNet. Our evaluation reveals a strategic trade off: TGEA maximizes success rates via global evolutionary exploration, while SGSA excels in efficiency through saliency-guided local refinement. Notably, SGSA outperforms baselines on adversarially trained models, effectively exploiting the perceptually aligned gradients of robust networks. PriSM demonstrates that strategic exploitation of surrogate information matched to search algorithm characteristics substantially improves query efficiency in black-box robustness evaluation.

## Broader Impact Statement

This work develops query-efficient adversarial attacks for black-box settings. Our primary motivation is to evaluate and ultimately improve the robustness of Deep Neural Networks (DNNs) deployed in safety-critical domains such as medical diagnostics and autonomous systems. We recognize that the techniques we study could, in principle, be misused to evade deployed models with fewer queries, particularly against production API-based services (e.g., cloud ML platforms, content moderation systems) where existing rate limiting may prove insufficient. We explicitly do not endorse such misuse. To mitigate these risks, we recommend multi-layered defenses including: (1) query pattern detection to identify structured exploration, (2) response randomization to reduce information leakage per query, (3) adaptive rate limiting based on suspicious activity, and (4) ensemble diversity to reduce cross-model transferability (Tang et al., 2024). Systematically probing systems for failure modes through responsible red-teaming is a cornerstone of security practice. By demonstrating that surrogate priors substantially reduce query costs, our results highlight the urgent need for defenses that reduce the transferability of decision-boundary geometry and saliency patterns. PriSM enables developers to stress-test models against sophisticated, resource-constrained adversaries, identify weaknesses before deployment, and iteratively build more robust learning systems.

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

## A    Hyperparameter and Configuration Details

This appendix details the specific hyperparameter values used for the proposed adversarial attacks. Table 12 outlines the coefficients used in the fitness functions, while Table 13 lists the specific algorithmic settings.

We explicitly verify the impact of these hyperparameter choices in **Appendix B**. Please refer to the Ablation Studies therein for a detailed sensitivity analysis and validation of the configuration values listed above.

Table 12: Fitness Function Coefficients. We distinguish between Target Optimization parameters ($\alpha_t, \gamma_t$) and Surrogate Initialization parameters ($\alpha_s, \gamma_s$) to highlight their distinct roles in exploitation vs. exploration.

| Component | Parameter Symbol | Value |
|---|---|---|
| **Target Opt.** | $\alpha_t$ (Max Incorrect) | 0.4 |
| | $\gamma_t$ (True Class Penalty) | 1.1 |
| **Surrogate (SEGI)** | $\alpha_s$ (Max Incorrect) | 0.3 |
| | $\gamma_s$ (True Class Penalty) | 1.3 |
| | $\delta$ (Centroid Distance) | 0.9 |
| | Diversity Multiplier | 10 |

**Weight Asymmetry.** Supported by Appendix B2, we employ asymmetric weights to balance exploration and exploitation. SEGI uses a higher penalty ($\gamma_s = 1.3$) and lower targeted weight ($\alpha_s = 0.3$) to enforce diversity, driving candidates away from the true class without converging to a single *easiest* incorrect class. In contrast, the target stage employs balanced weights ($\gamma_t = 1.1, \alpha_t = 0.4$) to facilitate stable convergence into missclassification.

Table 13: Algorithmic Hyperparameters.

| Component | Parameter | Value |
|---|---|---|
| **SEGI (GA)** | Population Size | 60 |
| | Parents Mating | 5 |
| | Mutation Rate | 10% |
| | Top Solutions $\rightarrow$ CMA-ES | 13 |
| | Injection Noise ($\sigma$) | 0.035 |
| **TGEA (CMA)** | Population Size | 16 |
| | Max Iterations | 90 |
| | Active CMA Update | True |
| **SGSA** | Saliency Scale Factor | 10.0 |
| | Gaussian Sigma ($\sigma$) | 1.0 |
| | Update Frequency | 50 iters |
| | Fallback Threshold | 100 iters |
| | Random Fallback Prob. | 5% |
| | Multi-scale Scales | [1.0, 0.5, 0.25] |
| | Temporal Decay ($\lambda$) | 0.9 |
| | Attention Weight ($\beta$) | 0.3 |
| **General** | Max Query Budget | 1000 |

## B  Ablation Studies

To validate the contributions of key components in PriSM, we perform targeted ablation studies on both TGEA and SGSA. All ablations use the same evaluation protocol with CIFAR-10 and ImageNet datasets at $\epsilon = 0.1$ unless otherwise specified. Best results per metric are highlighted with gray shading.

### B.1  Transfer-Guided Evolutionary Attack (TGEA)

#### B.1.1  Analysis of Surrogate Fitness Function

We ablate individual terms of SEGI's fitness function: $\alpha_s$, $\gamma_s$, and $\delta$. Results appear in Table 14.

Table 14: SEGI fitness ablation on CIFAR-10 ($\epsilon = 0.1$).

| | InceptionV3 | | VGG16 | |
|---|---|---|---|---|
| Variant | ASR $\uparrow$ | AQ $\downarrow$ | ASR $\downarrow$ | AQ $\downarrow$ |
| Full Function | **91.43** | **105.26** | **89.80** | 127.82 |
| $\alpha_s = 0$ | 91.37 | 110.77 | 89.48 | 139.58 |
| $\gamma_s = 0$ | 90.18 | 107.11 | 86.93 | 136.84 |
| $\delta = 0$ | 88.39 | 97.10 | 87.30 | **124.15** |

Table 15: SEGI fitness ablation on ImageNet ($\epsilon = 0.1$).

| | DenseNet | | EfficientNet | |
|---|---|---|---|---|
| Variant | ASR $\uparrow$ | AQ $\downarrow$ | ASR $\uparrow$ | AQ $\downarrow$ |
| Full Function | **89.47** | 111.25 | **89.10** | 148.42 |
| $\alpha_s = 0$ | 83.83 | 92.79 | 83.81 | **133.13** |
| $\gamma_s = 0$ | 75.40 | **80.89** | 82.54 | 138.31 |
| $\delta = 0$ | 85.19 | 112.37 | 86.79 | 160.83 |

As expected, in the untargetted attack scenario, the $\gamma_s$ term proves most critical, its removal ($\gamma_s = 0$) causes the largest ASR drops across all models (up to 14% on DenseNet). This validates the importance of aggressively suppressing the true class during surrogate evolution. The $alpha_s$ and $delta$ terms provide smaller but consistent improvements, with occasional query efficiency gains when removed at the cost of reduced ASR.

In addition to evaluating the impact of removing these terms entirely, we conducted a grid search to optimize their specific magnitudes. We varied $\alpha_s$, $\gamma_s$, and $\delta$ across the range of values $\alpha_s \in \{0.8, 1.5\}$, $\gamma_s \in \{0.5, 1.0\}$ and $\delta \in \{0.5, 1.5\}$ to verify that our chosen asymmetric configuration provides a good balance between exploration and exploitation compared to other feasible settings.

The analysis reveals that the default configuration outperforms the grid-searched variants across all models. Notably, increasing $\alpha_s$ to 0.8 or 1.5 (which aggressively rewards incorrect class confidence) leads to a slight degradation in ASR and increased query costs. This validates our intuition. The surrogate evolution stage (SEGI) should prioritize **exploration** via a lower $\alpha_s$ and higher $\gamma_s$, ensuring a diverse population of candi-

Table 16: SEGI fitness grid search (InceptionV3 & VGG16).

| | InceptionV3 | | VGG16 | |
|---|---|---|---|---|
| Params | ASR ↑ | AQ ↓ | ASR ↑ | AQ ↓ |
| $\alpha_s = 0.3, \gamma_s = 1.3, \delta = 0.9$ (Default) | **91.43** | **105.26** | **89.80** | **127.82** |
| $\alpha_s = 0.8, \gamma_s = 1.0, \delta = 1.5$ | 88.88 | 122.95 | 83.02 | 165.03 |
| $\alpha_s = 0.8, \gamma_s = 1.0, \delta = 0.5$ | 89.15 | 118.40 | 83.25 | 158.90 |
| $\alpha_s = 0.8, \gamma_s = 0.5, \delta = 1.5$ | 85.04 | 127.20 | 84.12 | 184.75 |
| $\alpha_s = 0.8, \gamma_s = 0.5, \delta = 0.5$ | 86.45 | 121.10 | 83.50 | 178.40 |
| $\alpha_s = 1.5, \gamma_s = 0.5, \delta = 0.5$ | 87.50 | 119.80 | 83.80 | 180.20 |
| $\alpha_s = 1.5, \gamma_s = 0.5, \delta = 1.5$ | 86.90 | 125.50 | 83.65 | 188.10 |
| $\alpha_s = 1.5, \gamma_s = 1.0, \delta = 0.5$ | 89.99 | 115.23 | 83.92 | 175.73 |
| $\alpha_s = 1.5, \gamma_s = 1.0, \delta = 1.5$ | 88.91 | 120.31 | 82.12 | 172.16 |

Table 17: SEGI fitness grid search (DenseNet & EfficientNet).

| | DenseNet | | EfficientNet | |
|---|---|---|---|---|
| Params | ASR ↑ | AQ ↓ | ASR ↑ | AQ ↓ |
| $\alpha_s = 0.3, \gamma_s = 1.3, \delta = 0.9$ (Default) | **89.47** | **111.25** | **89.10** | **148.42** |
| $\alpha_s = 0.8, \gamma_s = 1.0, \delta = 1.5$ | 87.34 | 127.37 | 86.83 | 178.97 |
| $\alpha_s = 0.8, \gamma_s = 1.0, \delta = 0.5$ | 88.10 | 124.10 | 87.50 | 169.20 |
| $\alpha_s = 0.8, \gamma_s = 0.5, \delta = 1.5$ | 86.22 | 128.52 | 85.88 | 178.12 |
| $\alpha_s = 0.8, \gamma_s = 0.5, \delta = 0.5$ | 86.90 | 125.60 | 86.15 | 172.50 |
| $\alpha_s = 1.5, \gamma_s = 1.0, \delta = 0.5$ | 89.05 | 123.83 | 89.01 | 163.09 |
| $\alpha_s = 1.5, \gamma_s = 0.5, \delta = 0.5$ | 88.40 | 126.90 | 88.20 | 167.40 |
| $\alpha_s = 1.5, \gamma_s = 0.5, \delta = 1.5$ | 86.80 | 130.50 | 86.40 | 176.80 |
| $\alpha_s = 1.5, \gamma_s = 1.0, \delta = 1.5$ | 87.32 | 129.32 | 87.15 | 178.5 |

dates rather than greedily optimizing for a specific incorrect label, which is better handled by the subsequent CMA-ES phase.

### B.1.2 Analysis of Target Fitness Function

To determine the optimal balance for the target fitness function, we performed a grid search over the hyperparameters $\alpha_t \in \{0.1, 1.0, 1.5\}$ and $\gamma_t \in \{0.5, 1.0, 1.6\}$. We compare these variations against the default settings, results are presented in Table 18.

Table 18: Target fitness grid search (InceptionV3 & VGG16).

| | InceptionV3 | | VGG16 | |
|---|---|---|---|---|
| Params | ASR ↑ | AQ ↓ | ASR ↑ | AQ ↓ |
| $\alpha_t = 0.4, \gamma_t = 1.1$ (Default) | **91.43** | **105.26** | **89.80** | **127.82** |
| $\alpha_t = 0.1, \gamma_t = 1.0$ | 88.39 | 119.22 | 83.89 | 176.01 |
| $\alpha_t = 1.0, \gamma_t = 1.0$ | 89.09 | 115.69 | 84.64 | 173.22 |
| $\alpha_t = 1.5, \gamma_t = 0.5$ | 89.08 | 138.19 | 83.85 | 175.08 |
| $\alpha_t = 0.1, \gamma_t = 0.5$ | 88.75 | 119.85 | 84.15 | 175.50 |
| $\alpha_t = 1.5, \gamma_t = 1.6$ | 89.12 | 119.95 | 84.72 | 173.10 |
| $\alpha_t = 0.1, \gamma_t = 1.6$ | 87.95 | 116.51 | 83.20 | 177.50 |
| $\alpha_t = 1.0, \gamma_t = 1.6$ | 88.85 | 119.43 | 84.35 | 174.40 |
| $\alpha_t = 1.0, \gamma_t = 0.5$ | 89.05 | 136.32 | 84.22 | 174.80 |
| $\alpha_t = 1.5, \gamma_t = 1.0$ | 89.15 | 121.59 | 84.50 | 173.90 |

Table 19: Target fitness grid search (DenseNet & EfficientNet).

| | DenseNet | | EfficientNet | |
|---|---|---|---|---|
| Params | ASR ↑ | AQ ↓ | ASR ↑ | AQ ↓ |
| $\alpha_t = 0.4, \gamma_t = 1.1$ (Default) | 89.47 | 111.25 | 89.10 | 148.42 |
| $\alpha_t = 0.1, \gamma_t = 1.0$ | 85.42 | 105.73 | 81.82 | 135.44 |
| $\alpha_t = 1.0, \gamma_t = 1.0$ | 93.15 | 132.10 | 89.17 | 147.05 |
| $\alpha_t = 1.5, \gamma_t = 0.5$ | 80.73 | 105.38 | 88.11 | 165.29 |
| $\alpha_t = 0.1, \gamma_t = 0.5$ | 88.20 | 114.50 | 84.50 | 140.10 |
| $\alpha_t = 1.5, \gamma_t = 1.6$ | **93.22** | 132.80 | **89.25** | 148.50 |
| $\alpha_t = 0.1, \gamma_t = 1.6$ | 82.50 | **102.10** | 78.40 | **129.80** |
| $\alpha_t = 1.0, \gamma_t = 1.6$ | 90.80 | 124.60 | 87.65 | 142.30 |
| $\alpha_t = 1.0, \gamma_t = 0.5$ | 86.40 | 116.20 | 88.65 | 158.40 |
| $\alpha_t = 1.5, \gamma_t = 1.0$ | 89.50 | 122.90 | 88.90 | 152.10 |

The grid search results highlight distinct behaviors across architectures. For **InceptionV3** and **VGG16**, the default settings ($\alpha_t = 0.4, \gamma_t = 1.1$) remain superior, achieving the highest ASR and lowest query counts. This suggests these models benefit from a balanced approach where the target class is encouraged moderately.

In contrast, the more complex architectures, **DenseNet** and **EfficientNet**, perform better with higher scaling factors ($\alpha_t \geq 1.0$). DenseNet specifically achieves a +3.7% boost in ASR (93.22%) when using $\alpha_t = 1.5, \gamma_t = 1.6$ compared to the default. This indicates that for highly connected or optimized networks, the attack requires stronger gradients towards the target class to successfully guide the optimization out of local minima.

### B.1.3 Analysis of Population Size

Table 20 evaluates CMA-ES population sizes: 6 (small), 16 (default), and 26 (large).

On CIFAR-10, smaller populations excel in both metrics, suggesting SEGI's initialization enables rapid convergence on simpler datasets. On ImageNet, the default size achieves highest ASR while small populations remain most query-efficient, indicating a complexity dependent trade off where larger search spaces benefit from moderate population sizes to avoid premature convergence.

Table 20: SEGI population size on CIFAR-10 ($\epsilon = 0.1$).

| Pop. Size | InceptionV3 | | VGG16 | |
|---|---|---|---|---|
| | ASR ↑ | AQ ↓ | ASR ↑ | AQ ↓ |
| Small (6) | **92.99** | **90.27** | **92.22** | **117.48** |
| Default (16) | 91.43 | 105.26 | 89.80 | 127.82 |
| Large (26) | 84.70 | 128.02 | 81.51 | 161.28 |

Table 21: SEGI population size on ImageNet ($\epsilon = 0.1$).

| Pop. Size | DenseNet | | EfficientNet | |
|---|---|---|---|---|
| | ASR ↑ | AQ ↓ | ASR ↑ | AQ ↓ |
| Small (6) | 87.04 | **94.90** | 87.45 | **126.60** |
| Default (16) | **89.47** | 111.25 | **89.10** | 148.42 |
| Large (26) | 85.23 | 176.10 | 86.40 | 134.93 |

### B.1.4 Analysis of Computational Overhead

### B.2 Saliency-Guided Square Attack (SGSA)

### B.2.1 Analysis of Guidance Components

We evaluate two ablations: **Two-Scale** variant using scales [1.0, 0.5] and **Single-Scale** (original resolution saliency only). Table 22 shows results.

Table 22: SGSA guidance ablation on CIFAR-10 ($\epsilon = 0.1$).

| Variant | InceptionV3 | | VGG16 | |
|---|---|---|---|---|
| | ASR ↑ | AQ ↓ | ASR ↑ | AQ ↓ |
| Default | **85.64** | **53.47** | **82.85** | 77.62 |
| Two-Scale | 84.77 | 58.80 | 81.80 | **68.89** |
| Single-Scale | 79.06 | 57.60 | 78.12 | 77.81 |

Table 23: SGSA guidance ablation on ImageNet ($\epsilon = 0.1$).

| Variant | DenseNet | | EfficientNet | |
|---|---|---|---|---|
| | ASR ↑ | AQ ↓ | ASR ↑ | AQ ↓ |
| Default | 93.99 | **55.85** | 86.99 | 89.56 |
| Two-Scale | 92.81 | 76.34 | 87.11 | 82.02 |
| Single-Scale | **94.25** | 59.84 | 85.86 | 98.63 |

Single-scale saliency underperforms across metrics (e.g., +10% queries on EfficientNet). The two-scale variant shows competitive performance, but the default three-scale configuration achieves the best overall balance, validating that multi-scale aggregation captures transferable structural patterns at multiple feature hierarchies.

### B.2.2 Analysis of Saliency Scale Factor

We compare scale factors 2, 5, and 10 (default). Results in Table 24.

Table 24: SGSA scale factor on CIFAR-10 ($\epsilon = 0.1$).

| Scale | InceptionV3 | | VGG16 | |
|---|---|---|---|---|
| | ASR ↑ | AQ ↓ | ASR ↑ | AQ ↓ |
| Default (10) | **85.64** | **53.47** | **82.85** | 77.62 |
| Factor 5 | 79.50 | 57.53 | 78.39 | 73.59 |
| Factor 2 | 80.00 | 57.48 | 77.86 | **69.31** |

Table 25: SGSA scale factor on ImageNet ($\epsilon = 0.1$).

| Scale | DenseNet | | EfficientNet | |
|---|---|---|---|---|
| | ASR ↑ | AQ ↓ | ASR ↑ | AQ ↓ |
| Default (10) | **93.99** | **55.85** | **86.99** | **89.56** |
| Factor 5 | 93.75 | 59.20 | 86.17 | 92.29 |
| Factor 2 | 93.95 | 59.32 | 85.24 | 91.53 |

Lower scale factors occasionally reduce queries (e.g., factor 2 on VGG16) but consistently harm ASR. The default factor of 10 achieves highest or near highest ASR universally, demonstrating that stronger smoothing generates more stable and transferable guidance maps.

### B.2.3 Analysis of saliency update frequency

We evaluate the sensitivity of the attack to the saliency update frequency $k$, comparing the default ($k = 50$) against more frequent ($k \in \{10, 25\}$) and lazier ($k = 100$) schedules. This parameter controls the trade-off between using fresh gradient information and the computational overhead of updating the surrogate priors.

The results indicate that the default update frequency ($k = 50$) provides a trade-off between the freshness of the surrogate priors and the computational overhead of updating them. While lazier updates ($k = 100$) can

Table 26: Update frequency $k$ on CIFAR-10.

| | InceptionV3 | | VGG16 | |
|---|---|---|---|---|
| $k$ Value | ASR ↑ | AQ ↓ | ASR ↑ | AQ ↓ |
| 10 | 85.16 | 68.83 | 81.97 | 72.59 |
| 25 | 85.94 | 53.55 | 81.56 | 70.72 |
| **50 (Default)** | 85.64 | **53.47** | **82.85** | 77.62 |
| 100 | **86.28** | 56.30 | 82.35 | **67.49** |

Table 27: Update frequency $k$ on ImageNet.

| | DenseNet | | EfficientNet | |
|---|---|---|---|---|
| $k$ Value | ASR ↑ | AQ ↓ | ASR ↑ | AQ ↓ |
| 10 | 93.09 | 62.62 | 86.25 | **73.75** |
| 25 | 92.50 | 90.07 | **87.12** | 101.04 |
| **50 (Default)** | **93.99** | **55.85** | 86.99 | 89.56 |
| 100 | 92.81 | 62.07 | 83.71 | 117.46 |

occasionally improve efficiency on simpler landscapes, they risk working with old gradient information on more complex decision boundaries, leading to increased query costs. Overly frequent updates ($k = 10$) yield diminishing returns, often increasing computational cost without significant gains in attack success rate.

### B.2.4 Analysis of attention weight

We analyze the impact of the attention guidance weight $\beta$, which controls the influence of the surrogate's attention map on the perturbation direction. We compare the default $\beta = 0.3$ against weaker ($\beta = 0.1$), stronger ($\beta \in \{0.5, 0.75\}$) guidance, and no attention ($\beta = 0$).

Table 28: Attention weight $\beta$ on CIFAR-10.

| | InceptionV3 | | VGG16 | |
|---|---|---|---|---|
| $\beta$ Value | ASR ↑ | AQ ↓ | ASR ↑ | AQ ↓ |
| 0 (No Attention) | 79.06 | 59.02 | 78.72 | **73.30** |
| 0.1 | 84.90 | 50.58 | 82.81 | 74.05 |
| **0.3 (Default)** | **85.64** | 53.47 | **82.85** | 77.62 |
| 0.5 | 85.36 | **46.97** | 81.25 | 61.32 |
| 0.75 | 85.41 | 56.73 | 81.77 | 64.51 |

Table 29: Attention weight $\beta$ on ImageNet.

| | DenseNet | | EfficientNet | |
|---|---|---|---|---|
| $\beta$ Value | ASR ↑ | AQ ↓ | ASR ↑ | AQ ↓ |
| 0 (No Attention) | 93.85 | 58.75 | 85.57 | 92.30 |
| 0.1 | 93.42 | 58.82 | 88.02 | 89.49 |
| **0.3 (Default)** | **93.99** | **55.85** | 86.99 | **89.56** |
| 0.5 | 93.17 | 61.77 | 86.11 | 101.86 |
| 0.75 | 93.30 | 58.39 | **88.02** | 90.00 |

Removing attention ($\beta = 0$) degrades ASR significantly on CIFAR-10 ($-6.6\%$ on InceptionV3), confirming that attention maps provide complementary guidance beyond gradient-based saliency. While increasing attention guidance can improve query efficiency and ASR on specific architectures, the default $\beta = 0.3$ provides the most stable performance across all models, particularly for DenseNet.

### B.2.5 Analysis of Temporal Decay

We evaluate the sensitivity of the temporal mechanism to the decay factor $\lambda$, which determines how much historical gradient information is retained. We compare the default $\lambda = 0.9$ against lower values representing faster decay.

Table 30: Temporal decay $\lambda$ on CIFAR-10.

| | InceptionV3 | | VGG16 | |
|---|---|---|---|---|
| $\lambda$ Value | ASR ↑ | AQ ↓ | ASR ↑ | AQ ↓ |
| 0.1 | 84.38 | 61.22 | 81.25 | 69.05 |
| 0.5 | 83.04 | 57.41 | 80.80 | **67.99** |
| 0.75 | 85.58 | **53.10** | 82.20 | 71.82 |
| **0.9 (Default)** | **85.64** | 53.47 | **82.85** | 77.62 |

Table 31: Temporal decay $\lambda$ on ImageNet.

| | DenseNet | | EfficientNet | |
|---|---|---|---|---|
| $\lambda$ Value | ASR ↑ | AQ ↓ | ASR ↑ | AQ ↓ |
| 0.1 | 94.12 | 62.83 | **88.54** | 90.27 |
| 0.5 | **94.49** | 61.80 | 87.50 | 89.70 |
| 0.75 | 93.33 | **53.74** | 87.00 | 92.69 |
| **0.9 (Default)** | 93.99 | 55.85 | 86.99 | **89.56** |

The results suggest that a higher decay factor ($\lambda \approx 0.9$) is generally efficient, balancing ASR and query efficiency. While lower values (e.g., $\lambda = 0.1$) occasionally yield marginal ASR gains (as seen in EfficientNet), they often result in higher query costs or reduced stability on other architectures (e.g., VGG16), justifying the choice of $\lambda = 0.9$ to effectively smooth the gradient estimation over time.

### B.3 Computational Overhead

**Saliency-Guided Square Attack (SGSA)**

To quantify the cost of surrogate guidance, we measured the wall clock time of SGSA against the baseline Square Attack on 1000 samples. Table 32 summarizes the computational overhead introduced by multi-scale saliency map generation and guidance operations.

Table 32: Computational overhead of SGSA saliency guidance.

| Dataset | Total Time (s) | Pure Search (s) | Saliency Overhead |
|---------|----------------|-----------------|-------------------|
| CIFAR-10 | 1.85 | 1.70 | 0.15 (8.1%) |
| ImageNet | 5.23 | 5.12 | 0.11 (2.16%) |

The saliency overhead remains minimal, accounting for only 8.1% on CIFAR-10 and 2.16% on ImageNet. Crucially, this overhead is amortized across the entire attack budget. Since the guidance substantially reduces the total number of queries needed, the net wall clock time decreases despite the per-iteration cost, confirming that surrogate guidance provides practical computational efficiency.

**Transfer-Guided Evolutionary Attack (TGEA)**

To quantify the computational cost of transfer-guided evolutionary optimization, we measured the wall clock time of TGEA's components on 1000 samples. Tables 33 and 34 show the breakdown for each variant.

Table 33: TGEA-SEGI computational profile (seconds per attack).

| Dataset | Random Population Time (s) | Total Time (s) | Population Init (s) | Attack (s) |
|---------|---------------------------|----------------|---------------------|------------|
| CIFAR-10 | 3.29 | 6.34 | 4.24 | 2.10 |
| ImageNet | 19.36 | 48.68 | 32.53 | 16.14 |

Table 34: TGEA-TASI computational profile (seconds per attack).

| Dataset | Total Time (s) | Population Init (s) | Attack (s) |
|---------|----------------|---------------------|------------|
| CIFAR-10 | 5.34 | 3.21 | 2.13 |
| ImageNet | 22.27 | 6.80 | 15.47 |

TGEA introduces computational overhead for transfer-based population initialization, which accounts for 67% of total time for SEGI on CIFAR-10 and 67% for SEGI on ImageNet. For TASI, initialization overhead is lower at 60% on CIFAR-10 and 15% on ImageNet.

Overall, the overhead of the attacks must be evaluated in context: both attacks achieve substantially higher attack success rates while using fewer queries. In real-world settings where query budgets are strictly limited (e.g., production APIs with rate limiting, detection systems, service costs), the ability to succeed with fewer queries is often more valuable than raw wall-clock time. The modest computational investment (8% for SGSA, 1.9-2.5× for TGEA) enables attacks that would otherwise fail entirely within the query budget, making both methods highly practical for security evaluation and adversarial robustness testing.

### B.4 Robustness to Surrogate Model Selection

To assess PriSM's sensitivity to surrogate choice, we evaluate SGSA and SEGI using GoogLeNet and DenseNet as surrogates (vs. default ResNet18), attacking InceptionV3 and VGG16 on CIFAR-10 at $\epsilon = 0.1$. Results in Table 35.

Table 35: Impact of surrogate model on SGSA and SEGI performance (CIFAR-10, $\epsilon = 0.1$).

| Method | Surrogate | InceptionV3 | | VGG16 | |
|---|---|---|---|---|---|
| | | ASR ↑ | AQ ↓ | ASR ↑ | AQ ↓ |
| **SGSA** | ResNet18 (default) | **85.64** | **53.47** | 82.85 | **77.62** |
| | GoogLeNet | 85.99 | 53.66 | 82.33 | 75.03 |
| | DenseNet | 85.41 | 52.27 | **83.05** | 70.67 |
| **SEGI** | ResNet18 (default) | **91.43** | **105.26** | **89.80** | **127.82** |
| | GoogLeNet | 88.73 | 121.38 | 84.69 | 167.73 |
| | DenseNet | 88.48 | 119.55 | 86.38 | 151.74 |

Both methods exhibit robustness across surrogate architectures, with SGSA demonstrating particularly stable performance. SGSA maintains consistently high ASR and low query counts regardless of surrogate choice, with DenseNet achieving the lowest query count on InceptionV3 (52.27). This stability is remarkable given the architectural diversity tested, SGSA's query efficiency varies by less than 50% across surrogates, whereas traditional transfer attacks can see 2-3× degradation with mismatched architectures. SEGI shows slightly more sensitivity with 3-5% ASR variation, though ResNet18 performs best overall. Notably, performance degradation remains modest even with architecturally dissimilar surrogates (e.g., GoogLeNet's depthwise-separable convolutions vs. VGG16's standard convolutions), validating that multi-scale saliency patterns transfer robustly across diverse architectures. The consistency suggests that readily available pretrained models can be used as surrogates without substantial performance loss, with SGSA being especially resilient to surrogate selection.

## C  Attack Convergence Analysis

To provide a deeper insight into the trade off between query efficiency and attack success rate, we visualize the convergence behavior of SGSA and TGEA against baseline methods. The following plots illustrate the Attack Success Rate (ASR) as a function of the number of queries.

### C.1  MNIST Models

Figure 7 shows the convergence on MNIST for both $\epsilon = 0.2$ and $\epsilon = 0.3$.

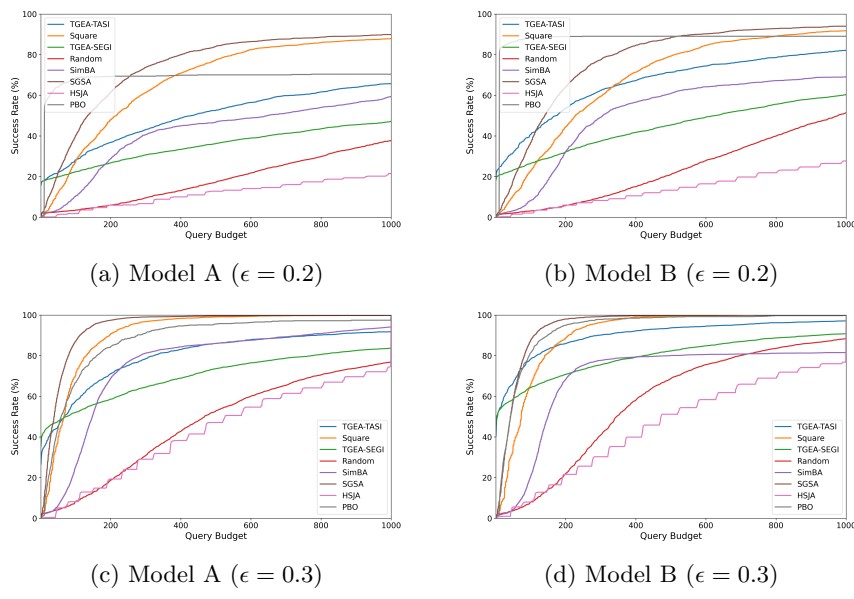

(a) Model A ($\epsilon = 0.2$)   (b) Model B ($\epsilon = 0.2$)

(c) Model A ($\epsilon = 0.3$)   (d) Model B ($\epsilon = 0.3$)

Figure 7: **MNIST Models:** Convergence analysis for $\epsilon = 0.2$ and $\epsilon = 0.3$.

## C.2 CIFAR-10 Standard Models

Figure 8 compares performance on InceptionV3 and VGG16.

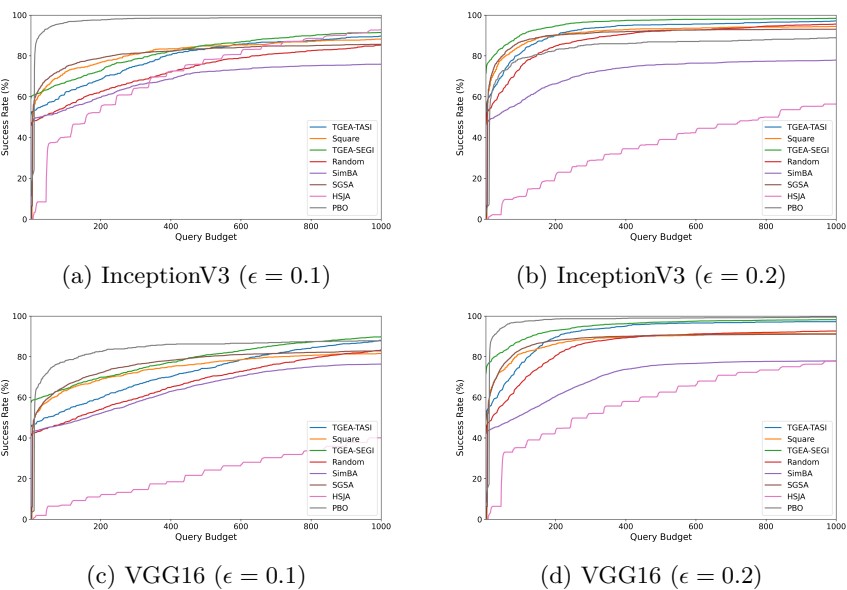

(a) InceptionV3 ($\epsilon = 0.1$)

(b) InceptionV3 ($\epsilon = 0.2$)

(c) VGG16 ($\epsilon = 0.1$)

(d) VGG16 ($\epsilon = 0.2$)

Figure 8: **CIFAR-10 Standard Models:** ASR vs. Average Queries.

## C.3 CIFAR-10 Robust Models

Figure 9 displays results on adversarially trained networks. Notably, on the SOTA WRN-94-16, SGSA matches or exceeds the success rate of baselines while maintaining superior query efficiency.

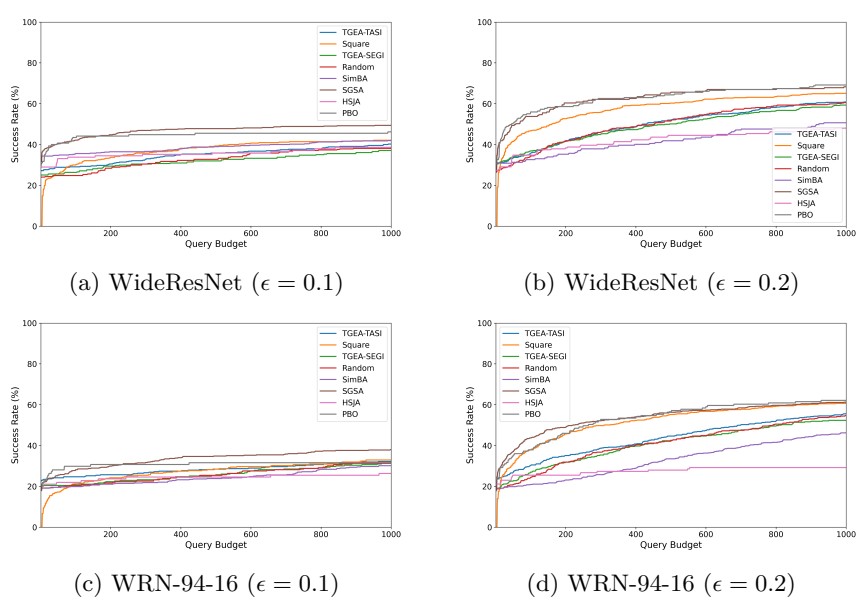

(a) WideResNet ($\epsilon = 0.1$)

(b) WideResNet ($\epsilon = 0.2$)

(c) WRN-94-16 ($\epsilon = 0.1$)

(d) WRN-94-16 ($\epsilon = 0.2$)

Figure 9: **CIFAR-10 Robust Models:** ASR vs. Average Queries.

## C.4    ImageNet Models

Figure 10 illustrates performance on high resolution ImageNet models. SGSA (brown) consistently demonstrates steeper convergence curves in the early query phase.

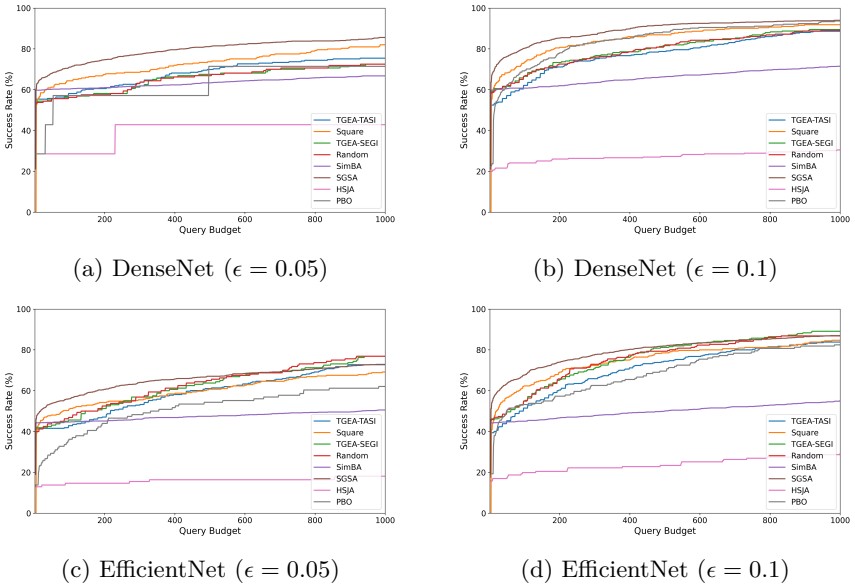

(a) DenseNet ($\epsilon = 0.05$)    (b) DenseNet ($\epsilon = 0.1$)

(c) EfficientNet ($\epsilon = 0.05$)    (d) EfficientNet ($\epsilon = 0.1$)

Figure 10: **ImageNet Models:** ASR vs. Average Queries.

