# OpenReview forum: "PriSM: Prior-Guided Search Methods for Query Efficient Black-Box Attacks"
_TMLR — Accepted by TMLR_

### Review · Reviewer_z2u9 · 2025-12-31

**Summary Of Contributions:**

The paper proposes the PriSM method to balance the query efficiency and attack success rate under black-box attack constraints.

The main contributions include
(1) Transfer-guided evolutionary attack (TGEA), which initializes CMA-ES (Covariance Matrix Adaptation Evolution Strategy) with the transfer-based priors of the surrogate model, with
TASI (Transfer-Attack Seeded Initialization) for generating diverse initial candidates from multiple types of attacks to the surrogate model;
And SEGI (Surrogate-Evolved Genetic Initialization) for evolving a high-quality initial population.

(2) Saliency-guided Square Attack (SGSA), which uses multi-scale gradient maps from a surrogate model to guide perturbation placement and sizing.

(3) Evaluation on standard benchmarks demonstrate the effectiveness of the method.

**Audience:**

Yes

**Audience Explanation:**

The paper is highly relevant to TMLR's readership.

(1) Adversarial robustness is an important topic in secure and reliable machine learning.

(2) Query-efficient black-box attacks are important for real-world applications.

(3) The paper demonstrates a new way to balance query efficiency and attack success rate, with extensive evaluations and ablations.

**Broader Impact Concerns:**

The paper does not present a broader impact statement, yet the method (the topic of black-box attacks) may have potential misuse and societal implications.

**Claims And Evidence:**

Yes

**Claims Explanation:**

The major claims are well-supported by extensive experiments, ablation studies, and intuitive justifications:

(1) the strategic trade off: TGEA improving ASR and SGSA reducing query costs are consistently supported by Tables 2-9;

(2) Competitive with state-of-the-art hybrid attacks is supported by the comparisons with PBO;

(3) Robustness to surrogate choice is supported by Table 21, Appendix B.3, where SGSA and SEGI tend to maintain stable performance across three CNN-based surrogate models.

**Requested Changes:**

(1) Please clarify the baseline choices. The latest method for comparison is PBO, which was published in 2024. Please justify why the newly proposed methods are not involved. Please also justify the selected defense strategies.

(2) While Table 10 of Appendix A lists all hyperparameters, a discussion of the sensitivity would strengthen the reproducibility.

(3) Please add a discussion of potential misuse and societal implications.

---

### Review · Reviewer_KPep · 2026-01-04

**Summary Of Contributions:**

The paper proposes **PriSM (Prior-Guided Search Method)**, a novel framework that incorporates adversarial transferability to enhance black-box adversarial search. Specifically, PriSM exploits two types of transferable surrogate information: decision boundary geometry and loss landscape topology.

The authors further demonstrate the importance of these transferable priors by introducing two complementary attacks:
(1) **TGEA (Transfer-Guided Evolutionary Attack)**, which leverages boundary information, and
(2) **SGSA  (Saliency-Guided Square Attack)**, which leverages loss topography.

Finally, the authors evaluate their attacks on the MNIST, CIFAR-10, and ImageNet datasets. The results demonstrate the effectiveness of the proposed methods.

**Audience:**

Yes

**Audience Explanation:**

The reviewer strongly believes that the TMLR's audience would be very much interested  in the findings of this paper.

**Broader Impact Concerns:**

None.

**Claims And Evidence:**

Yes

**Claims Explanation:**

To the best of the reviewer's knowledge, the claims made by the authors are accurate, convincing, and clearly presented. The justification provided for the proposed TGEA and SGSA methods appears to be sound.

**Requested Changes:**

As black-box adversarial searches are notoriously computationally expensive, the reviewer is concerned about the additional computational overhead that the TGEA and SGSA methods may incur. The authors provide only vague and minimal information in the Appendix regarding the computational costs of their methods. While the proposed approaches improve attack success rates, it remains unclear why and how they would be practical in real-world settings, given that incorporating adversarial transferability into black-box searches may further increase computational overhead.

The reviewer recommends that the authors include one or more tables reporting the time required to generate successful adversarial examples using TGEA and SGSA, and compare these times to those of the selected baselines. These experiments could be conducted using CIFAR-10 and/or ImageNet.

---

> ### Author Response · Authors · 2026-01-17
> **Response to Reviewer KPep**
>
> We sincerely thank the reviewer for their thoughtful evaluation and positive assessment of our work. We are grateful for the recognition that our claims are accurate, convincing, and clearly presented, and that our justification for TGEA and SGSA is sound. We appreciate the reviewer's concern regarding computational overhead, which is indeed a critical practical consideration for black-box attacks. We addressed this concern by conducting comprehensive timing experiments for CIFAR-10 and ImageNet. The detailed results are now presented in **Section B.3 "Computational Overhead"** in the Appendix, which include:
>
> 1. Complete timing breakdowns for TGEA-SEGI and TGEA-TASI showing population initialization time, attack execution time, total time per attack, and direct baseline comparisons against random population (Tables 33 and 34).
> 2. SGSA overhead analysis comparing total time, pure search time, and saliency computation overhead (Table 32).
> 3. Analysis of the overhead in context of real-world constraints.
>
> Our results show that:
>
> **TGEA**: Introduces a 1.9-2.5× wall-clock overhead compared to baselines:
> - TGEA-SEGI: 6.34s on CIFAR-10 (vs 3.29s random baseline, 1.9× overhead), 48.68s on ImageNet (vs 19.36s random baseline, 2.5× overhead)
> - TGEA-TASI: 5.34s on CIFAR-10, 22.27s on ImageNet
>
> **SGSA**: Introduces very minimal overhead:
> - CIFAR-10: 1.85s total (vs 1.70s pure search), 0.15s saliency overhead (8.1%)
> - ImageNet: 5.23s total (vs 5.12s pure search), 0.11s saliency overhead (2.16%)
>
> Critically, this overhead must be evaluated in context: both attacks achieves substantially higher attack success rates (e.g., 15-35% improvement in ASR) while using fewer queries. In real-world settings where query budgets are strictly limited (e.g., production APIs with rate limiting), the ability to succeed with fewer queries is often more valuable than raw wall clock time. The increase in per-attack time enables adversarial examples that would otherwise fail entirely within the query budget, making both methods highly practical for security evaluation and adversarial robustness testing.

---

### Review · Reviewer_xB8V · 2026-01-05

**Summary Of Contributions:**

The paper proposes Prior-guided Search Methods, PriSM, a framework for improving query efficiency in black-box adversarial attacks by systematically exploiting transferable priors obtained from surrogate models. The key idea is to distinguish different types of transferable information and to match them with proper search strategies. The paper introduces two attack methods: Transfer-Guided Evolutionary Attack (TGEA) and Saliency-Guided Square Attack (SGSA). TGEA uses surrogate-informed initialization to warm-start CMA-ES, and SGSA incorporates multi-scale saliency information from a surrogate model into Square Attack. Experiments on the image benchmark datasets show improved reductions in query usage, while maintaining competitive attack success rates.

[Strengths]

* The paper presents a decomposition of transferable priors and argues that different priors should be exploited by different optimization methods.
* The focus on query efficiency in the adversarial black-box setting is timely and practically relevant.

[Weaknesses]

* A large number of hyperparameters are introduced, yet the empirical analysis of their sensitivity is limited.

* The conceptual relationship between TGEA and SGSA within a single unified framework remains somewhat unclear. The methods currently appear more like parallel cases studies than tightly integrated components.

* SGSA relies heavily on Square Attack and its implicit CNN inductive bias, raising concerns about generalization to non-convolutional architectures such as vision transformers.

**Audience:**

Yes

**Audience Explanation:**

The problem of query-efficient black-box adversarial attacks is of clear interest to the community, particularly researchers working on robustness, security, and black-box optimization. The idea of explicitly categorizing transferable priors and aligning them with specific search mechanisms offers a useful perspective that goes beyond incremental algorithmic tweaks. Moreover, the paper’s focus on practical query budgets and its evaluation on standard benchmarks suggest that its findings would be relevant to both academic and applied audiences.

**Broader Impact Concerns:**

The proposed methods lower the query cost of black-box adversarial attacks, which could increase the risk of practical misuse against deployed systems with limited query budgets. While the paper acknowledges defensive motivations, the Broader Impact discussion could be strengthened by more concretely addressing realistic attack scenarios (e.g., API-based services) and by discussing mitigation strategies such as rate limiting, response randomization, or attack detection mechanisms.

**Claims And Evidence:**

No

**Claims Explanation:**

The empirical results partially support the claim that surrogate-guided priors can reduce query counts in black-box attacks across several datasets and models.

However, several central claims are not fully supported by convincing by convincing evidence. In particular, although the paper emphasizes the robustness and general applicability of its methods, systematic sensitivity analyses of the many introduced hyperparameters are largely absent from the main paper. The appendix mainly contains ablation studies that toggle components on or off, rather than exploring performance trends across a range of parameter values. This makes it difficult to assess tuning cost, stability, and reproducibility.

**Requested Changes:**

[Major]

* The paper introduces many hyperparameters, yet their influence is not systematically analyzed.I strongly believe that the main manuscript should include sesitivity analysis of key parameters over meaningful ranges, rather than relegating limited ablations to the appendix. In particular, it would be important to analyze how optimal hyperparameters choices vary across datasets and target models.

* While the paper argues that different priors call for different exploitation strategies, the unifying principle of PriSM remains underdeveloped. The authors could more clearly articulate: 1) why specific priors are matched to specific search algorithms, 2) whether the two methods are complementary or mutually exclusive, and 3) how a practitioner should choose between them under different constraints.

* Since SGSA is fundamentally based on Square Attack and spatial saliency, and the paper itself notes the connection to CNN inductive bias, it is critical to test SGSA on transformer-based models (e.g., ViT). If SGSA performs poorly, the paper need to analyze why and discuss possible adaptations.


[Minor]

* The citation style appears inconsistent with common conventions. Currently, nearly all citations place author names explicitly in the main text. (may need to change citet to citep in latex)

* In Section 3.2, the definition of the "Project" function used in the Square Attack formulation appears to be missing.

* Across several figures, the font size of text inside the figures is too small to be easily readable.

* While Figure 1 provides an overall intuition for surrogate-based initialization, it does not clearly explain why TGEA is expected to move toward a global optimum. Strengthening the intuition would help readers better understand why improved initialization leads to performance gains.

* It is unclear whether the parameters $\alpha$, $\gamma$ used in the surrogate-specific fitness function are intended to be identical to those used in the target model fitness function. If they are the same, a justification would be helpful.

* The motivation for TGEA appears to be closely related to margin-based considerations. However, the fitness function assigns different weights to the two terms in the margin, and the rationale for this asymmetry is not clearly explained. Providing theoretical or empirical justification for using different weights would improve the interpretability of the method.

---

> ### Author Response · Authors · 2026-01-17
> **Response to Reviewer xB8V [1/3]**
>
> We sincerely thank the reviewer for their thorough and constructive evaluation of our work. We are grateful for the recognition that our focus on query efficiency in black-box adversarial attacks is `"timely and practically relevant"`, and that our decomposition of transferable priors offers a useful perspective `"beyond incremental algorithmic tweaks"`. We particularly appreciate the acknowledgment that our findings would be of clear interest to both academic and applied audiences in the robustness and security community. We also thank the reviewer for identifying important areas where our work could be strengthened. We have carefully addressed each concern through substantial revisions and additional experiments, which we detail below, with all changes marked in red.
>
> ### **Major Changes**
>
> **Major Change 1: Comprehensive Sensitivity Analysis**
>
> We thank the reviewer for this constructive criticism. We fully agree that a systematic analysis of hyperparameters is essential to demonstrate the method's robustness and reproducibility. We have significantly expanded the main manuscript to include a more comprehensive sensitivity analysis in Section 5.3 (Sensitivity Analysis and Component Validation), rather than only relegating these findings to the appendix. We also performed extensive grid searches and ablations across different datasets and target architectures to characterize the influence of key parameters:
>
> - For TGEA (Figure 5): We performed a grid search over meaningful ranges for both the Surrogate Fitness and Target Fitness functions, specifically analyzing their interplay.
> - For SGSA (Figure 6): We conducted a systematic sweep of key hyperparameters, including the Attention Weight ($\beta$), Temporal Decay ($\lambda$), Saliency Scale Factor, and Update Frequency ($k$). Furthermore, we expanded our component analysis with a two-scale ablation.
>
> These analyses demonstrate that our methods exhibit stable performance across reasonable parameter ranges, and provide evidence-based results for practitioners on parameter selection across different datasets and models.

---

> ### Author Response · Authors · 2026-01-17
> **Response to Reviewer xB8V [2/3]**
>
> **Major Change 2: Clarifying the Unifying Framework**
>
> We thank the reviewer for his valuable feedback and appreciate the opportunity to articulate the unifying principle of PriSM: **structural alignment**, where the nature of the surrogate prior matches the operational mechanics of the black-box algorithm.
>
> #### **(1). Methodological Justification**
>
> We acknowledge the reviewer's concern regarding our algorithm prior pairing logic. Our decisions were driven by operational compatibility, matching each prior's information structure to the algorithmic mechanisms that can efficiently exploit it. More broadly, our goal is to demonstrate how different types of transferable priors *naturally* align with specific optimization paradigms: population-level geometric constraints for evolutionary strategies (TGEA) vs spatial topographical cues for local search methods (SGSA).
>
> **TGEA** (Boundary Geometry + CMA-ES): CMA-ES operates by adapting a multivariate normal distribution $\mathcal{N}(\mu, \Sigma)$. It fundamentally requires population level data to estimate a mean vector ($\mu$) and covariance matrix ($\Sigma$). The boundary geometry derived from surrogates aims to provide precisely this: a population of diverse points near the decision boundary. Due to the empirical transferability of decision boundaries across models trained on similar data [1], initializing CMA-ES with boundary-proximal populations positions the search closer to the target model's adversarial regions, reducing the queries needed to find successful perturbations (we empirically observe 30-60% reduction vs random initialization). While other population based algorithms (e.g., Differential Evolution, Genetic Algorithms) could also potentially benefit from boundary geometry initialization, we selected CMA-ES for its adaptive covariance mechanism and established effectiveness in black-box attacks [2]. Point based methods cannot leverage this population structure.
>
> **SGSA** (Loss Topography + Square Attack): Square Attack relies on discrete spatial decisions: (1) *where* to place a square perturbation, and (2) *how large* to make it. Saliency maps provide spatial gradient magnitudes, which map directly to these decisions. We use saliency to guide probabilistic location sampling (Eq. 20) and adaptive size adjustment. Coordinate-wise methods (e.g., SimBA) or decision-based attacks (e.g., HSJA) lack the spatial mechanisms to utilize this 2D topographical guidance
>
> [1] Papernot et al., "The Space of Transferable Adversarial Examples" arxiv 2017.
>
> [2] Ilyas et al., "Black-box adversarial attacks using Evolution Strategies" arxiv 2021.
>
> We acknowledge that our original framing may have been too assertive given the scope of our empirical evaluation. We have revised section 4.1 to present structural alignment as a design consideration that guided our methods and proved effective in practice, while being clear that we demonstrate this principle through two instantiations rather than establish it as a generic theory.
>
> #### **(2). Independence of Methods**
> We also clarify the relationship between our methods to avoid confusion. TGEA and SGSA are *operationally* independent; they are distinct algorithms rather than sequential steps in a single pipeline. However, they can be seen as "*strategically* complementary" because they target distinct transferable properties that align with different search scales. We have revised the manuscript to explicitly articulate this distinction.
>
> #### **(3).  Practitioner Guidance**
> Based on our experimental results, we provide the following evidence-based guidance for practitioners. For applications with a **strict query budget**, we recommend **SGSA**, which averaged fewer queries than baseline methods. However, for scenarios where **high attack success rate** (ASR) is the priority, **TGEA-SEGI** is the better choice, achieving the higher ASR (e.g., 98.36% on InceptionV3). In the context of **adversarial training**, **SGSA** demonstrated superior robustness against defenses (e.g., on WRN-94-16). Finally, for **simple datasets** (e.g., MNIST), both methods are highly effective, with SGSA offering a slight advantage in query efficiency and ASR.
>
> In the revision, we have added section 6.4 (Practitioner Guidance) to further guide the readers.

---

> ### Author Response · Authors · 2026-01-17
> **Response to Reviewer xB8V [3/3]**
>
> **Major Change 3: Evaluation on Vision Transformers**
>
> We thank the reviewer for this critical suggestion. We agree that verifying generalization beyond CNNs is important, particularly given SGSA's reliance on spatial saliency priors which align well with the inductive bias of convolutions. To address this, we extended our evaluation to include two transformer-based architectures: **ViT-B/16** and **DeiT-S**.
>
> Contrary to the concern that SGSA might fail due to the lack of CNN-like feature locality in transformers, the method performed remarkably well. On DeiT-S ($\epsilon=0.1$): SGSA achieved the highest ASR (87.90%) and lowest query count (104.52), significantly outperforming the baselines. On ViT-B/16: SGSA maintained superior query efficiency, compared to baselines, demonstraiting it remains a viable strategy for transformer architectures. These results suggest that while ViTs process information globally via self-attention, the critical decision features (and thus the optimal perturbation locations) remain spatially localized around the object of interest. In the revised version we have added tables 10, 11 with the results of these experiments to detail these findings.
>
> ### **Minor Changes**
>
> **Minor change 1**: Fixed citation style inconsistencies throughout the manuscript (changed \citet to \citep where appropriate).
>
> **Minor change 2**: Added the definition of the Project function in Section 3.2 (Equation 3), which ensures the adversarial example remains within the valid ϵ-ball bound.
>
> **Minor change 3**: Increased font sizes in key figures for improved readability.
>
> **Minor change 4**: TGEA's population-based initialization provides CMA-ES with both a good starting location near the target's decision boundary and directional information via the population's covariance structure. This covariance reveals effective perturbation directions, allowing CMA-ES to intelligently search. Single point methods are inherently "greedy" and might get trapped by the non-transferable, local optima in black-box landscapes. TGEA's population, in contrast to single point methods, aims to capture the global geometry of the decision boundary allowing the optimizer to potentially step over local minima and converge to better solutions. As requested, in the revised version, we have added this geometric intuition.
>
> **Minor change 5**: We clarify that these parameters are indeed distinct and optimized for different objectives. To resolve this confusion, we have updated the notation to explicitly distinguish between surrogate parameters ($\alpha_s, \gamma_s$) and target parameters ($\alpha_t, \gamma_t$). As detailed in our new ablation studies (Tables 16, 17, 18 & 19), the surrogate phase benefits from asymmetric weights to prioritize **exploration**, while the target phase requires more balanced weights for stable **exploitation**. We have revised the notation throughout the manuscript and added the comparative grid search results to justify their distinct values.
>
> **Minor change 6**: We employ asymmetric weights to explicitly manage the **exploration/ exploitation trade off**. The intuition is that in the surrogate initialization phase, a higher true class penalty ($\alpha_s$) and lower target weight ($\gamma_s$) prioritize **exploration** more, ensuring the population covers a diverse range of potential directions rather than potentially only focusing just on a "easy" target. As mentioned above, to validate this emperically, we added grid searches demonstrating that asymmetric configurations are often more optimal for attack success rates.
>
> **Minor Change 7 (Broader Impact)**: Strengthened the Broader Impact Statement by concretely addressing realistic attack scenarios (API-based services with rate limiting) and discussing specific mitigation strategies including query pattern detection, response randomization, adaptive rate limiting, ensemble diversity, and boundary obfuscation.

---

### Author Response · Authors · 2026-01-17
**Summary of Changes**

We thank the reviewers for their time and the effort they took to provide insightful feedback! Your feedback has helped us identify a few ways to improve the paper:

**1. Comprehensive Sensitivity Analysis (Reviewers `xB8V` and `z2u9`)**: Added Section 5.3 and more ablations with extensive grid search experiments analyzing key hyperparameters for both TGEA and SGSA. Results demonstrate stable performance across parameter ranges and provide evidence-based guidance for practitioners.

**2. Computational Overhead Analysis (Reviewer `KPep`)**: Added Section B.3 with detailed timing experiments on CIFAR-10 and ImageNet (Tables 32-34), showing TGEA introduces 1.9-2.5× overhead while SGSA adds only 2-8% overhead, justified by substantially improved attack success rates and query efficiency in real-world constrained settings.

**3. Vision Transformer Evaluation (Reviewer `xB8V`)**: Extended evaluation to vision transformer architectures **ViT-B/16** and **DeiT-S** (Tables 10-11), demonstrating SGSA's effectiveness on transformer architectures despite lack of CNN-like inductive bias.

**4. Clarified Unifying Framework (Reviewer `xB8V`)**: Strengthened articulation of PriSM's "structural alignment" principle in Section 4.1, clarified operational independence of methods, and added Section 6.4 with practitioner guidance.

**5. Broader Impact Statement (Reviewers `xB8V` and `z2u9`)**: Added comprehensive statement addressing realistic attack scenarios (API-based services), potential misuse concerns, and concrete mitigation strategies including query pattern detection, response randomization, and adaptive rate limiting.

**6. Minor Improvements (Reviewers `xB8V` and `z2u9`)**: Fixed citation inconsistencies, added missing function definition, increased figure font sizes, enhanced Figure 1 intuition, distinguished surrogate/target parameters with empirical validation, and clarified baseline/defense choices.

All changes are marked in red in the revised version. We believe these revisions comprehensively address all reviewer concerns and substantially strengthen the paper. We are happy to provide any additional clarifications or experiments as needed.

---

### Decision · Action_Editor_fRBK · 2026-02-25

**Recommendation:** Accept as is

**Audience:**

Yes

**Audience Explanation:**

The topic of this paper belongs to adversarial machine learning, so there would be many individuals in TMLR's audience who are interested in knowing the findings of this paper.

**Claims And Evidence:**

Yes

**Claims Explanation:**

This paper introduces PriSM, a framework that improves black-box adversarial attacks by using two types of information from surrogate models: decision boundary geometry and loss landscape topography. The evidence supporting these claims comes from extensive experiments on MNIST, CIFAR-10, and ImageNet, showing 30–60% query reductions over uninformed baselines while maintaining competitive success rates.